# Improvement of near-surface wind speed modeling through refined aerodynamic roughness length in high-roughness surface regions: implementation and validation in the Weather Research and Forecasting (WRF) model version 4.0

Jiamin Wang[1], Kun Yang[1,2], Jiarui Liu[1], Xu Zhou[3], Xiaogang Ma[4], Wenjun Tang[3], Ling Yuan[5], Zuhuan Ren[1]

[1]Ministry of Education Key Laboratory for Earth System Modeling, Department of Earth System Science, Institute for Global Change Studies, Tsinghua University, Beijing 100084, China.
[2]Renewables Research Center of Huairou Laboratory, Beijing 101499, China.
[3]National Tibetan Plateau Data Center, State Key Laboratory of Tibetan Plateau Earth System, Environment and Resources, Institute of Tibetan Plateau Research, Chinese Academy of Sciences, Beijing 100101, China.
[4] National Institute of Natural Hazards, Ministry of Emergency Management of China, Beijing, 100085, China.
[5]China State Shipbuilding Corporation Haizhuang Windpower Co., Ltd., Chongqing 401123, China.

*Correspondence to*: Kun Yang (yangk@tsinghua.edu.cn)

**Abstract.** Aerodynamic roughness length ($z_0$) is a key parameter determining near-surface wind profiles, significantly influencing wind-related studies and applications. In high-roughness surface areas, surface roughness has been substantially altered by land use changes such as urbanization. However, many numerical models still assign long-standing and fixed $z_0$ based on traditional land cover types, neither accounting for shifts in land cover nor updating class-specific $z_0$, leaving $z_0$ values in high-roughness surface regions outdated and unreliable. To address this issue, this study proposed a cost-effective method to estimate $z_0$ values at weather stations by adjusting $z_0$ values to minimize the wind speed differences between ERA5 reanalysis data and weather station observation data. Using this approach, $z_0$ values were derived for 1,837 stations in the high-roughness surface areas across China. Based on these estimates, a high-resolution monthly gridded $z_0$ dataset was then developed for high-roughness surface areas in China using Random Forest Regression algorithm. Simulations with Weather Research and Forecasting (WRF) model show that implementation of the new $z_0$ dataset significantly improves the accuracy of 10-m wind speed over high-roughness surface areas, reducing mean wind speed errors by 79.8% and 78.0% compared to the default $z_0$ in WRF and a latest gridded $z_0$ dataset from Peng et al. (2022), respectively. Independent validations of 100-m wind speed against anemometer tower data further confirm the dataset's reliability. Therefore, this approach is valuable for wind-dependent studies and applications, such as urban planning, air quality management, and wind energy utilization, by enabling more accurate simulations of wind speed in high-roughness surface areas.

## 1 Introduction

With the rapid advancement of urbanization and industrialization, human activities and energy use are increasingly concentrated along the settlement-landscape continuum (Liu et al., 2014), particularly in high-roughness areas such as built-

up zones and inhabited vegetated landscapes. High-roughness surface regions not only significantly influence climate change
but also are highly sensitive to meteorological and climatic conditions (Kammen and Sunter, 2016). Among various
meteorological parameters, wind speed exerts great impacts on both environmental and human systems. One prominent
example is that wind speed is a crucial consideration for assessing the atmospheric pollutant dispersion capability (Manju et
al., 2002; Han et al., 2017). Specifically, mean flows and atmospheric turbulence are two key factors for pollutant removal
(Wong and Liu, 2013; Di Nicola et al., 2022). Also, wind speed regulates pollen dispersion and distribution that are
associated with public health (Roy et al., 2023). The utilization of wind energy in high-roughness surface areas also depends
on wind speed distribution (Ishugah et al., 2014; Stathopoulos et al., 2018; Tasneem et al., 2020). Proper utilization, through
measures such as suburban wind farms or building-integrated turbines, can minimize the need for transmission infrastructure.
Beyond energy considerations, wind speed characteristics play a critical role in design and planning of human settlements,
influencing both contemporary building practices (Hadavi and Pasdarshahri, 2020) and the preservation of historical-cultural
heritage (Li, Y. et al., 2023). Therefore, accurately characterizing wind speed is essential for guiding systematic regulation
and promoting sustainable development in high-roughness surface areas.
Aerodynamic roughness length ($z_0$) is a crucial parameter that determines near-surface wind speed profiles (Stull, 1988). As
a key input for atmospheric models, $z_0$ significantly influences wind speed-related applications, however, its representation
in existing numerical models often oversimplifies real-world conditions. Specifically, most models, such as the widely used
European Centre for Medium-Range Weather Forecasts Reanalysis v5 (ERA5), determine $z_0$ with long-standing and fixed
values based on traditional land cover types. Such treatment fails to reflect the impact of transitions between surface types
and changes in roughness elements within the same type, particularly the complexity of urban structures, thereby posing
significant challenges for accurate wind speed simulation and prediction over high-roughness surface areas (Wang et al.,
2024). Numerous studies have demonstrated that the changes of $z_0$, caused by land use changes, particularly urbanization
and industrialization, as well as deforestation and afforestation, significantly impacted wind speed. For instance, the increase
in $z_0$ has explained 70% of the wind speed reduction in Europe (Wever, 2012) and caused a 1.1 m/s decrease in eastern
China (Wu et al., 2018). Furthermore, Zhang et al. (2019) identified $z_0$ changes as a primary driver of long-term wind speed
trends in China, Europe, and North America. In line with these findings, Luu et al. (2023) showed that the rise in $z_0$, caused
by shifts from short vegetation to high vegetation and urbanization, partly contributes to the decline in mean and maximum
surface wind speed over Western Europe. A similar mechanism operated in Canada. At Sudbury Airport (Ontario), 10-m
wind speeds declined by ~34% during 1975-1995 mainly due to reforestation-induced increases in surface roughness
(Tanentzap et al., 2007). These findings highlight the need to refine $z_0$ in models by incorporating the effects of high-
roughness surface areas across urban-town settings and tall-vegetation landscapes. In addition to wind speed, $z_0$ also plays a
significant role in environmental processes. The difference in $z_0$ between urban and suburban areas is one of drivers causing
larger intensity of daytime urban heat islands in humid regions (Zhao et al., 2014; Li et al., 2019). Winckler et al. (2019)
showed that roughness changes are a primary control on deforestation's biogeophysical effects, notably surface temperature
responses. Therefore, accurate $z_0$ data in high-roughness surface areas can not only enhance the performance of atmospheric
numerical models, but also provide scientific support for formulating sustainable urban environmental management
strategies.
The estimation of $z_0$ in high-roughness surface areas traditionally relies on three primary approaches: the
micrometeorological method, the morphometric method, and a combination of these two methods. The micrometeorological
method, based on the Monin-Obukhov similarity theory (Monin and Obukhov, 1954), typically calculates $z_0$ using
observations from flux or anemometer towers (Grimmond et al., 1998; Liu et al., 2018). Although theoretically robust, this
method is limited by high costs of instruments and infrastructure (Grimmond and Oke, 1999), as well as the need for
homogeneous surface conditions (Wieringa, 1993; Bottema and Mestayer, 1998). The morphometric method usually
formulates mathematical models based on geometric characteristics and distribution density of high-roughness surface areas
(Raupach, 1992 and 1994; Bottema and Mestayer, 1998; Macdonald et al., 1998; Kanda et al.,2013; Shen et al., 2022; Shen
et al., 2024). However, these models often suffer from simplified assumptions and require high-resolution surface feature
data, which are costly to acquire (Grimmond and Oke, 1999; Zhang et al., 2017). The combination method, which
establishes a relationship between the $z_0$ ground truth obtained from micrometeorological method and high-resolution
surface feature data for regional-scale applications, has shown promise in specific regions, such as Tokyo and Nagoya
(Kanda et al., 2013), Beijing (Zhang et al., 2017), and Osaka subregions (Duan and Takemi, 2021). Nevertheless, the
limitations of the former two methods hinder its broader applications. Therefore, there is a considerable lack of reliable $z_0$
data in high-roughness surface regions.
To address the aforementioned challenges, this study proposed a low-cost method for estimating $z_0$ by integrating 10-m wind
speed at China Meteorological Administration (CMA) stations with 10-m wind speed and $z_0$ from ERA5 reanalysis data.
This approach takes advantage of the synergy between CMA's high-density station distribution and ERA5 reanalysis'
temporal continuity to substantially enhance the sample size of $z_0$ estimates. Based on these estimates, we have developed a
high-resolution monthly $z_0$ dataset for high-roughness surface areas in China using Random Forest Regression (RFR)
algorithm. The applicability of the new $z_0$ dataset have been assessed through its implementation in the Weather Research
and Forecasting (WRF) model for wind speed simulation. This study contributes to the advancement of mesoscale wind
speed simulation over high-roughness surface environments, which can promote wind field-dependent studies, such as urban
planning, wind energy utilization, and air quality management.
**2 Data and Method**
**2.1 Data**
In this study, we mainly utilized monthly gridded $z_0$ dataset from ERA5 (Hersbach et al., 2020 and 2023a), referred to as
$z_{0\_ERA5}$, along with hourly 10-m wind speed data from both ERA5 (Hersbach et al., 2023b) and surface weather station
observations provided by the China Meteorological Administration (CMA) during 2015-2019, to derive $z_0$ estimates at each
CMA station.
To extend the site-scale $z_0$ estimates into a gridded dataset at the regional scale, we applied the RFR algorithm, incorporating
six key features: variance of the slope ($\overline{\theta^2}$), terrain standard deviation within 0.01° window (*TSD*), percent tree cover (*PTC*),
leaf area index (*LAI*), normalized difference vegetation index (*NDVI*), and urban-rural classification (*URC*). $\overline{\theta^2}$ was derived
as an integral over orographic spectrum, capturing multi-scale orographic complexity with wave length from meter to 10 km
(Beljaars et al., 2004). We obtained $\overline{\theta^2}$ from the dataset accompanying the turbulent orographic form drag scheme in WRF
(Zhou et al., 2018), which was processed from the global 30″ GMTED2010 digital elevation model (Danielson & Gesch,
2011). *TSD* was calculated using elevation data from Shuttle Radar Topography Mission with a spatial resolution of 3
arcseconds (Jarvis et al., 2018). The *PTC* data were obtained from the MOD44B Version 6.1 Vegetation Continuous Fields
product (DiMiceli et al., 2022), which provides yearly data at a 250-m pixel resolution. The monthly 1-km *NDVI* data were
acquired from MOD13A3 product (Didan, 2021). The *LAI* data with an 8-day temporal interval and 500-m spatial resolution
were sourced from Yuan et al. (2011) and Lin et al. (2023). *URC* data were extracted from a 1-km global human settlements
map, which categorizes the rural-urban continuum into 19 distinct types (Li, X. et al., 2022 and 2023). To generate a
monthly $z_0$ dataset at a spatial resolution of 0.01° × 0.01°, all input datasets were linearly interpolated or resampled to the
target resolution. *LAI* data were averaged monthly by assigning each 8-day interval to the closest month.
Additionally, to compare with the existed $z_0$ datasets, a latest $z_0$ dataset developed by Peng et al. (2022) (denoted as $z_{0\_Peng}$)
was used by integrating it into the WRF model for wind speed simulation. This dataset was generated by applying machine
learning techniques to integrate FLUXNET ground-based observations and MODIS remote sensing data. Moreover, 100-m
wind speed data from 589 anemometer towers in China were utilized for two critical purposes. First, the comparison between
tower observations and ERA5 100-m wind speed data (Hersbach et al., 2023b) was used to validate the feasibility of the
assumption in the $z_0$ estimation method. Second, tower data were used as independent validations to evaluate the impact of
refined $z_0$ on wind speed simulations. These anemometer towers cover varying periods between 2004 and 2022 with a
temporal resolution of 10 min.
**2.2 Method for deriving $z_0$ at CMA stations**
First, the theoretical basis for deriving $z_0$ at CMA stations is presented. In the framework of Monin-Obukhov similarity
theory (Monin and Obukhov, 1954), the neutral logarithmic wind profile can be expressed with Equation (1).
$$u_z = \frac{u_*}{k}\ln\left(\frac{z-d}{z_0}\right) \tag{1}$$

where $u_z$ is the wind speed (m/s) at height $z$, the measuring height above ground (m); $u_*$ is the friction velocity (m/s); $k$ is
the von Karman constant and equals to 0.4, and $d$ is the zero-plane displacement height (m), calculated as $d = 20/3\, z_0$ using
a widely accepted empirical formula (Watts et al., 2000).
Based on Equation (1), the 100-m neutral wind speed for ERA5 and CMA stations can be expressed in Equations (2) and (3),
respectively.
$$u_{100\_ERA5} = u_{10\_ERA5} \frac{\ln\left(\frac{100 - d_{ERA5}}{z_{0\_ERA5}}\right)}{\ln\left(\frac{10 - d_{ERA5}}{z_{0\_ERA5}}\right)} \tag{2}$$

$$u_{100\_CMA} = u_{10\_CMA} \frac{\ln\left(\frac{100 - d_{CMA}}{z_{0\_CMA}}\right)}{\ln\left(\frac{10 - d_{CMA}}{z_{0\_CMA}}\right)} \tag{3}$$

And then $z_0$ values at CMA stations can be estimated by the following three steps:
First, we assumed: (1) the near-surface wind speed difference between ERA5 and CMA is primarily attributed to $z_0$, and the
influence of $z_0$ diminishes with height. Consequently, the 100-m wind speed from ERA5 reanalysis is considered
comparable to that from observations; (2) the impact of atmospheric stability on wind speed is identical for both ERA5 and
CMA stations, allowing us to neglect stability correction terms under non-neutral conditions when deriving $z_0$ for each
hourly interval. The validity of these assumptions will be supported by the subsequent validation of wind speed simulations
based on the derived $z_0$ values (Section 3.3).
Second, we calculated the hourly $z_{0\_CMA}$ values based on Equations (2) and (3). Given that $u_{10\_ERA5}$, $u_{10\_CMA}$, and $z_{0\_ERA5}$
values are known, an optimal $z_{0\_CMA}$ value at each hour was derived through minimizing the difference between $u_{100\_ERA5}$
and $u_{100\_CMA}$ calculated using Equations (2) and (3). To align with Assumption (1), we only retained $z_{0\_CMA}$ values
corresponding to times when the percentage difference between the calculated $u_{100\_ERA5}$ and $u_{100\_CMA}$ was less than 10%.
ERA5 provides native 100-m winds, but here we use log-law–reconstructed 100-m winds from $u_{10\_ERA5}$ and $z_{0\_ERA5}$ instead.
The reason is that the $z_{0\_CMA}$ is derived under the assumption that stability-correction term is neglected. This means that the
100-m wind speeds in Equations (2) and (3) are both calculated without considering stability effects. However, the native
ERA5 100-m wind field inherently embeds model-diagnosed stability influences. Therefore, directly pairing native ERA5
100-m winds with our CMA log-law construction would amplify the error in the derived $\ln(z_0)$. In addition, the
reconstruction offers two practical advantages. First, it requires fewer variables and a more transparent linkage, relying only
on 10-m wind speeds and $z_0$ from reanalysis, together with 10-m wind speeds from observations; Second, our results
indicate that the $z_0$ estimates are not particularly sensitive to the choice of reference height (see Section 4. Discussion), so
there is no need to use native reanalysis winds at heights other than 10 m.
Third, these retained $z_{0\_CMA}$ values were grouped by months, and the monthly median values were selected as the final
roughness length ($z_{0\_optimal}$). To avoid unreasonable estimates, the values of $z_{0\_optimal}$ satisfying the condition that the
absolute difference between $\ln(z_{0\_optimal})$ and the corresponding $\ln(z_{0\_ERA5})$ does not exceed 2 were considered valid.
Finally, we obtained monthly $z_0$ estimates at 1,837 stations out of the 2,161 CMA stations.

## 2.3 Method for estimating gridded $z_0$ at regional scale

Machine learning serves as an effective tool for extending the $z_{0\_optimal}$ estimates at CMA stations to the regional scale. In this study, we employed the RFR algorithm (Equation (4)) (Breiman, 2001), a widely used method for similar applications (Duan and Takemi, 2021; Hu et al., 2022; Peng et al., 2022 and 2023). All samples were divided into training and test subsets at a ratio of 8:2 for each bin of $\ln(z_{0\_optimal})$, with the bins defined at intervals of 0.2. Sensitivity tests were conducted to determine the optimal number of decision trees in the RFR algorithm (Fig. 3b), resulting in the selection of 300 trees. The maximum depth of the trees was set to 18, and the minimum sample split was set to 5. Five-fold cross-validation shows the stable performance (Fig. 3d). Furthermore, the training and test results exhibit minimal sensitivity to the randomization seed used for dataset splitting (Fig. 3a). The resulting gridded aerodynamic roughness length data are referred to as $z_{0\_RFR}$.

$$\ln(z_0) = f\left(\overline{\theta^2}, TSD, PTC, LAI, NDVI, URC, month\right) \tag{4}$$

## 2.4 Model configuration

To demonstrate the applicability of gridded $z_{0\_RFR}$ data, the WRF (Version 4.0) Model (Skamarock et al., 2019) was used in this study to simulate wind speed with $z_{0\_RFR}$. For comparison, two additional simulations were performed: one utilized the WRF model's default roughness length ($z_{0\_Default}$) based on land cover types, and the other used $z_{0\_Peng}$.

First, we set $z_{0\_RFR}$ and $z_{0\_Peng}$ in WRF model, respectively. Given that $z_{0\_RFR}$ is concentrated in high-roughness surface areas, the missing values over other regions are filled with $z_{0\_Default}$. Notably, the setting of $z_{0\_Peng}$ in WRF is different from that of $z_{0\_RFR}$. In the WRF model, $z_0$ values over bare fraction and vegetated fraction are determined separately. Specifically, in the Noah-MP land surface model, $z_0$ is set to a constant over bare areas, while it is assigned by a look-up table according to vegetation type over vegetated areas. Peng et al. (2022) only provided the $z_0$ over vegetation areas, which is the gridded mean effective roughness length including vegetated fraction and bare fraction. Thus, before conducting the simulation of wind speed in the WRF model with the gridded $z_{0\_Peng}$, we adjusted the roughness length over vegetated fraction in each grid from $z_{0\_Peng}$. The specific adjustment of $z_{0\_Peng}$ in the WRF model is comprehensively described in the supplementary material Section 1. Apart from the difference in the sources of $z_0$, other model configurations for $z_{0\_RFR}$, $z_{0\_Default}$, and $z_{0\_Peng}$ are identical. The specific model configurations are as follows.

The simulation domains were configured with a "lat-lon" map projection, centered at coordinates 31.5°N, 109.0°E. As illustrated in Fig. 4b, nested domains were employed, with horizontal resolutions of 0.09° for Domain 1 (d01) and 0.03° for Domain 2 (d02). Specifically, d01 consisted of 225 grid points in the west-east direction and 191 in the south-north direction, while d02 consisted of 469 grid points in the west-east direction and 367 in the south-north direction. The vertical level had 70 layers and was stretched with dzstretch_s = 1.1 and dzstretch_u = 1.04. The model top was set to 50 hPa. The simulation periods spanned from March 31st to April 30th in 2019. The integral time interval was set to 30 seconds. The re-

initialization simulation was performed. Specifically, each simulation started at 12:00 local time (LT, LT=UTC+8) and ran
for 36 hours until 24:00 LT the next day. The first 12 hours were considered the spin-up time and the remaining hours were
used for analysis. Additionally, the initial and boundary conditions in the simulations were taken from hourly ERA5
reanalysis data, which provide pressure-level variables (geopotential height, air temperature, air humidity, and wind field)
(Hersbach et al., 2023c) and surface variables (surface air temperature, humidity, pressure, 10 m wind field, sea level
pressure, land surface temperature, soil temperature, and soil water content) (Hersbach et al., 2023b).
For physical parameterization schemes, the modified Thompson microphysics scheme (Thompson et al., 2008), Dudhia
scheme for shortwave radiation (Dudhia, 1989), Rapid Radiative Transfer Model (RRTM) scheme for longwave radiation
(Mlawer et al., 1997), Noah-MP land surface model (Niu et al., 2011), Yonsei University scheme for planetary boundary
layer (Hong et al., 2006), and Grell-Freitas for cumulus parameterization (Grell and Freitas, 2013) were adopted. The
cumulus parameterization scheme was activated only in the d01 domain and switched off in the d02 domain. A turbulent
orographic form drag scheme with description of the dynamic drag caused by sub-grid orography was also applied (Beljaars
et al., 2004; Zhou et al., 2018).

## 2.5 Calculation of statistical metrics

To evaluate the performance of the simulated wind speed with $z_{0\_RFR}$, $z_{0\_Default}$, and $z_{0\_Peng}$, three statistical metrics,
including correlation coefficient ($R$), mean absolute bias ($MAB$), and root mean square error ($RMSE$), were used in temporal
and spatial aspects. For the spatial performance assessment, the average 10-m wind speed simulation during April $1^{st}$ to $30^{th}$
in 2019 at each station was used to calculate $R$, $MAB$, and $RMSE$ with the CMA observations.
Regarding the temporal evaluation, the *index* (representing $R$, $MAB$, and $RMSE$) was calculated as the mean of the
corresponding metric for hourly 10-m wind speed during April $1^{st}$ to $30^{th}$ in 2019 across all CMA stations (Equation (5)).
$$index = \frac{\sum_{i=1}^{M} index_i}{M} \tag{5}$$

where $index_i$ denotes the respective metric value at the *i-th* station, and $M$ represents the total number of stations.
Additionally, to incorporate the direction of the bias into the wind-speed evaluation, we used the mean bias percentage ($MBP$)
to quantify the signed bias of ERA5 reanalysis and simulated wind speeds against observations from CMA stations and
anemometer towers (Equation (6)).
$$MBP = \frac{\bar{u}_{sim} - \bar{u}_{obs}}{\bar{u}_{obs}} \times 100\% \tag{6}$$

where $\bar{u}_{sim}$ represents mean wind speed from ERA5 or model simulations, and $\bar{u}_{obs}$ represents observed mean wind speed
from CMA stations and anemometer towers.
To more intuitively compare the performance of wind speed simulations using $z_{0\_Default}$, $z_{0\_Peng}$, and $z_{0\_RFR}$, we also
calculated the percentage reduction in wind speed error (*PRE*) achieved by $z_{0\_RFR}$ relative to $z_{0\_Default}$ and $z_{0\_Peng}$
(Equation (7)).

$$PRE = \frac{\left|\bar{u}_{z_{0\_*}} - \bar{u}_{observation}\right| - \left|\bar{u}_{z_{0\_RFR}} - \bar{u}_{observation}\right|}{\left|\bar{u}_{z_{0\_*}} - \bar{u}_{observation}\right|} \times 100\% \tag{7}$$

where $\bar{u}_{z_{0\_*}}$ represents $\bar{u}_{z_{0\_Default}}$ or $\bar{u}_{z_{0\_Peng}}$, and $\bar{u}$ denotes the mean 10-m or 100-m wind speed from simulations based on
$z_{0\_Default}$, $z_{0\_Peng}$, and $z_{0\_RFR}$, as well as from observations (CMA stations or anemometer towers).

## 3 Results

### 3.1 The distribution characteristics of the $z_0$ estimates at CMA stations

Figure 1a presents the spatial distribution of annual mean $z_{0\_optimal}$ values derived from 1,837 CMA stations, representing a
subset of all accessible 2,161 stations (Fig. S1a). These 1,837 stations are primarily located in the eastern, southern, and
central regions of China, with most stations having $z_0$ values ranging between 0.2 and 1.5 m. In contrast, the excluded 324
stations are mostly distributed in the western regions of China. The exclusions of these stations can be attributed to the poor
performance of ERA5 100-m wind speed data, which may result from altitude differences between the observation sites and
the model terrain, thereby rendering our initial assumption, i.e. ERA5 100-m wind speed data are reliable for $z_0$ estimation,
invalid in these areas. To test this, we evaluated the performance of ERA5 100-m wind speed by comparing it with 589
anemometer tower data, since CMA stations only provide 10-m wind speed observations. Overall, ERA5 shows a smaller
*MBP* in the eastern regions compared to the western regions (Fig. 2a). Therefore, the spatial distribution of the 1,837 stations
with valid $z_0$ values is reasonable.
Additionally, as a consistency check, we examined how the difference in $\ln(z_0)$ covaries with the 10-m wind-speed bias
between ERA5 reanalysis and station observations. Compared to the annual mean $\ln(z_{0\_optimal})$ derived from 1,837 stations,
the $\ln(z_{0\_ERA5})$ values are systematically lower at most locations, resulting in positive *MBP* values of 10-m wind speed
between ERA5 reanalysis data and station observations (Figs. 1b and 1c). The discrepancies between $\ln(z_{0\_ERA5})$ and
$\ln(z_{0\_optimal})$ are likely due to rapid urbanization around the majority of CMA stations, characterized by extensive
construction of buildings, which enhances surface roughness and consequently reduces near-surface wind speeds (Li et al.,
2018; Zhang and Wang, 2021). However, the impact of urbanization is likely not considered in the ERA5 reanalysis. Figures
2b and 2c depict the distribution of CMA stations classified by urban-rural categories. All stations are situated in high-
roughness surface areas, with the majority located in urban and town regions, highlighting the need to incorporate
urbanization effects into wind speed simulations to improve model accuracy. In contrast, at a few locations, where the
$\ln(z_{0\_ERA5})$ values are higher, the corresponding *MBP* values of 10-m wind speed are negative (Figs. 1b and 1c). The
influence of $\ln(z_0)$ difference on wind speed bias becomes more pronounced as the magnitude of $\ln(z_0)$ deviation increases
(Fig. 1d). Because $\ln(z_{0\_optimal})$ is defined as a monthly median of hourly $\ln(z_0)$, this cross-time statistic does not trivially
inherit the instantaneous relationship implied by Equations (1)-(3). The monotonic, theory-consistent pattern observed in the
binned $\ln(z_0)$ difference versus wind-speed *MBP* therefore serves as a post-aggregation consistency check, rather than as
proof. Accordingly, the robust consistency in the relationship between $z_0$ and wind speed preliminarily supports that
$z_{0\_optimal}$ is reasonable, and suggests that improving $z_0$ values over high-roughness surface areas in numerical models could
significantly enhance wind speed simulation accuracy. The validity of $z_{0\_optimal}$ will be assessed via independent validation
by comparing simulated wind speeds with observations (Section 3.3).

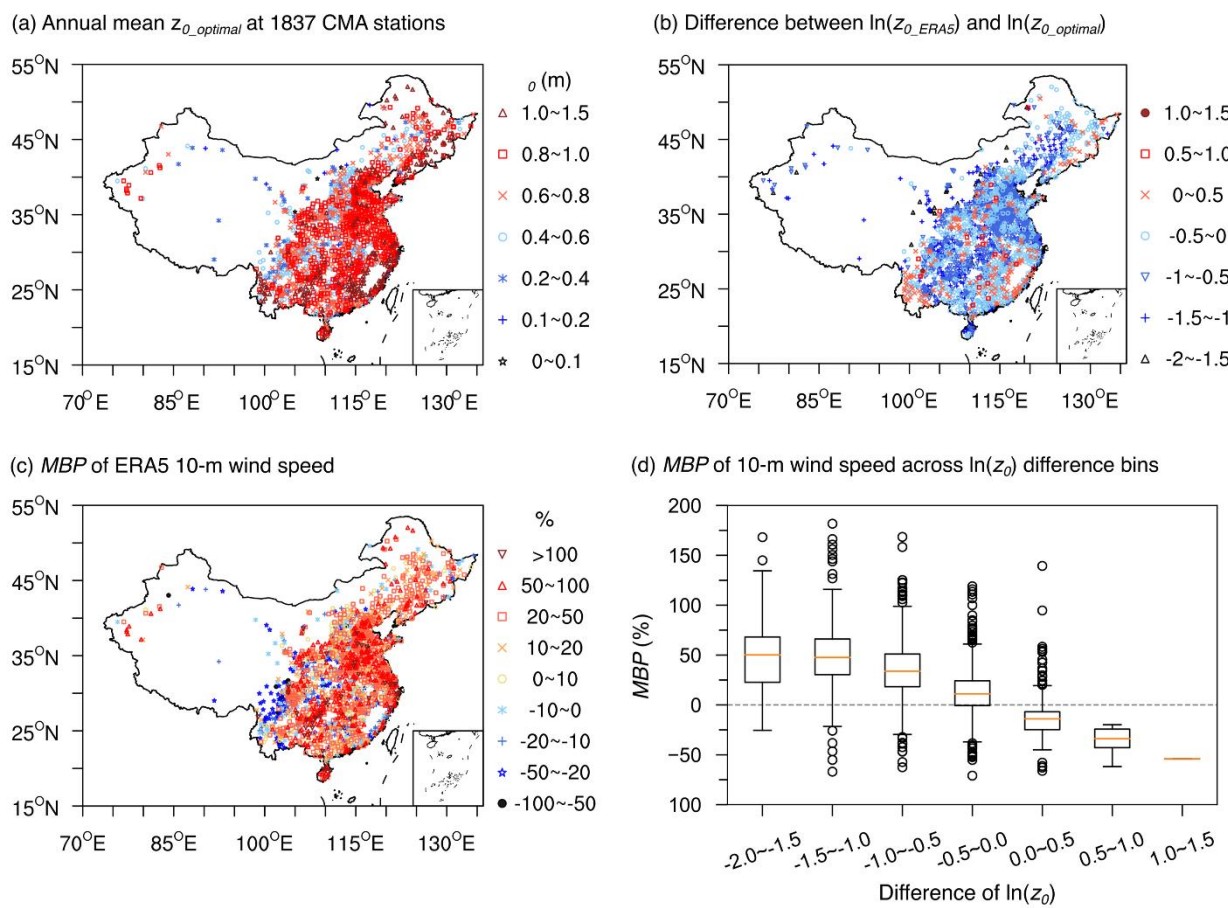

**Figure 1.** (a) Spatial distribution of annual mean $z_{0\_optimal}$ across 1,837 CMA stations. (b) Difference between annual mean $\ln(z_{0\_ERA5})$
and $\ln(z_{0\_optimal})$ (i.e., $\ln(z_{0\_ERA5})$ minus $\ln(z_{0\_optimal})$). (c) *MBP* of 10-m wind speed between ERA5 and CMA stations. (d) Boxplots
illustrating the statistical distribution of the *MBP* for 10-m wind speed shown in (c) across different intervals of $\ln(z_0)$ difference shown in
(b).

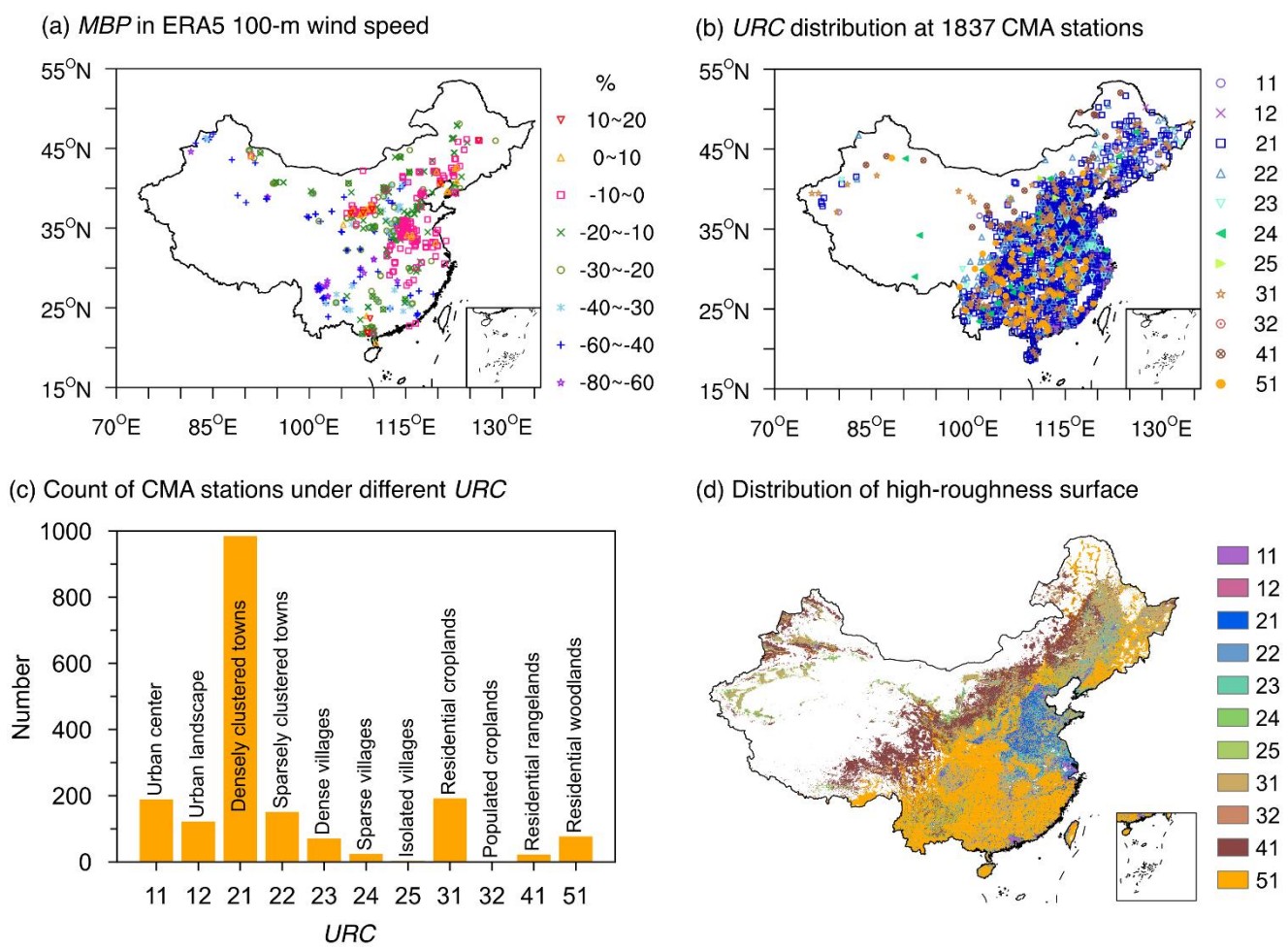

**Figure 2.** (a) *MBP* of 100-m wind speed between ERA5 and 589 anemometer towers. (b) Spatial distribution of urban-rural classification (*URC*) at 1,837 CMA stations. The legend on the right indicates the *URC* codes, with the corresponding *URC* types labeled in panel (c). (c) Number of CMA stations for each *URC*. The numerical labels on the x-axis represent the *URC* codes, with the specific *URC* types annotated on the bars. (d) Spatial distribution of high-roughness surface areas, which are composed of the 11 types covered by CMA stations in panel (b).

### 3.2 Development of a gridded $z_0$ dataset in high-roughness surface areas across China

To demonstrate the reliability and practicality of the estimated $z_{0\_optimal}$, we constructed a gridded $z_0$ dataset based on these estimations in order to apply it in numerical simulations. Given that the estimated $z_0$ values from 1,837 stations are located within high-roughness surface areas consisting of 11 distinct types (Figs. 2b and 2c), this study developed a monthly gridded $z_0$ dataset specifically for these categories of areas with a spatial resolution of $0.01° \times 0.01°$ using the RFR algorithm, referred to as $z_{0\_RFR}$. As a representative example, the $z_{0\_RFR}$ dataset was generated for the year 2019, and its spatial

coverage is shown in Fig. 2d. Although 2019 was chosen for demonstration, the RFR model itself is year-independent and
can be applied to other years, provided that the required input features are available. Six feature variables closely related to
$z_0$ were used as inputs, encompassing topographic characteristics ($\overline{\theta^2}$ and $TSD$), vegetation conditions ($PTC$, $LAI$, and $NDVI$),
and urban-rural distribution ($URC$).
Figure 3c shows that the RFR algorithm exhibits satisfactory performance on both training and test subsets. Feature
importance analysis reveals that topographic features and $PTC$ exert the most significant influence on $\ln(z_{0\_RFR})$ (Fig. 3e).
$z_0$ is primarily controlled by the characteristic height of surface roughness elements, particularly their relief. Consequently,
topographic features rank among the most influential factors. For vegetation-related features, $PTC$ not only reflects the
horizontal distribution of vegetation density but also serves as a proxy for the presence of tall roughness elements. By
contrast, $LAI$ mainly represents vegetation density, making it relatively less critical. Although $LAI$ is strongly correlated with
$NDVI$ ($R = 0.72$), its low importance is not driven by this collinearity. The $URC$ ranks only fourth in feature importance.
This ranking should not be interpreted as implying that land use or urbanization is insignificant. Rather, in our framework,
$URC$ is used mainly to delineate the study domain and to ensure that the RFR algorithm is applied only to high-roughness
surface areas. The aerodynamic effects of high-roughness elements, such as tall vegetation, buildings, and other
infrastructure, are already embedded in the wind observations from CMA stations. As a result, the influence of these
roughness elements is directly reflected in the $z_0$ values themselves, rather than being captured by the $URC$. Essentially,
$URC$ is not defined in terms of the morphological height and density of roughness elements; instead, it is derived from global
land-cover and population data (Li, X. et al., 2023), and is therefore weakly sensitive to $z_0$. For example, in categories of
Urban center and Urban landscape, there remains non-negligible tree cover, mean tree fractions of approximately 10% and
11%, respectively (Fig. S1b). This lowers $URC$'s ranking in feature-importance analyses. To better capture the influence of
roughness elements, more detailed surface parameters, such as building height and building density, would be helpful. Once
such data are widely accessible, they should be incorporated to further improve the accuracy of $z_0$ estimates.

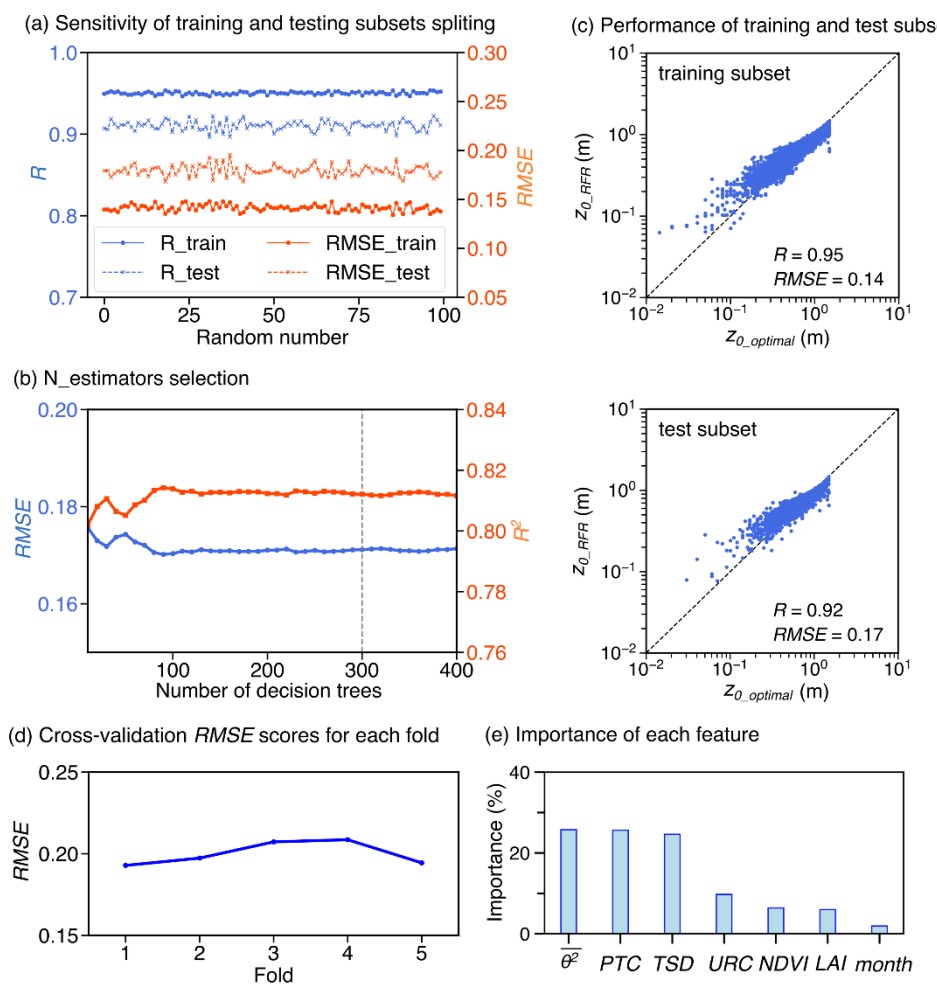

**Figure 3.** Sensitivity analysis and performance evaluation of the Random Forest Regression (RFR) algorithm. (a) Sensitivity of RFR results to the randomization seed for training and test subsets splitting. $R$ and $RMSE$ represent correlation coefficient and root mean square error, respectively. (b) Determination of the optimal number of decision trees. $R^2$ represents determination coefficient. (c) Performance of the RFR algorithm on the training and test subsets. The $R$ and $RMSE$ values are displayed. (d) Performance evaluation using five-fold cross-validation. (e) Importance scores of different feature variables.

The spatial distribution of $\ln(z_{0\_RFR})$ shows limited monthly variability (Fig. S2). The most pronounced monthly variations occur predominantly in the northern regions, likely because these areas exhibit strong seasonal changes in vegetation structure and biomass. The annual mean spatial distribution of $z_{0\_RFR}$, with values in high-roughness surface areas generally falling within the range of 0.3 to 0.9 m, exhibits distinct patterns compared to $z_{0\_Default}$ and $z_{0\_Peng}$ (Fig. 4a). In comparison with $z_{0\_Default}$ and $z_{0\_Peng}$, $z_{0\_RFR}$ shows a more homogeneous spatial distribution pattern across China. Specifically, in northern China, $z_{0\_RFR}$ values are consistently higher than those of both $z_{0\_Default}$ and $z_{0\_Peng}$, with $z_{0\_Default}$ generally

higher than $z_{0\_Peng}$. Conversely, in southern China, $z_{0\_Peng}$ values are significantly higher than both $z_{0\_Default}$ and $z_{0\_RFR}$.
However, in southeastern and southwestern China, $z_{0\_Default}$ values exceed those of $z_{0\_RFR}$, while in the remaining southern
areas, $z_{0\_RFR}$ maintains higher values compared to $z_{0\_Default}$.

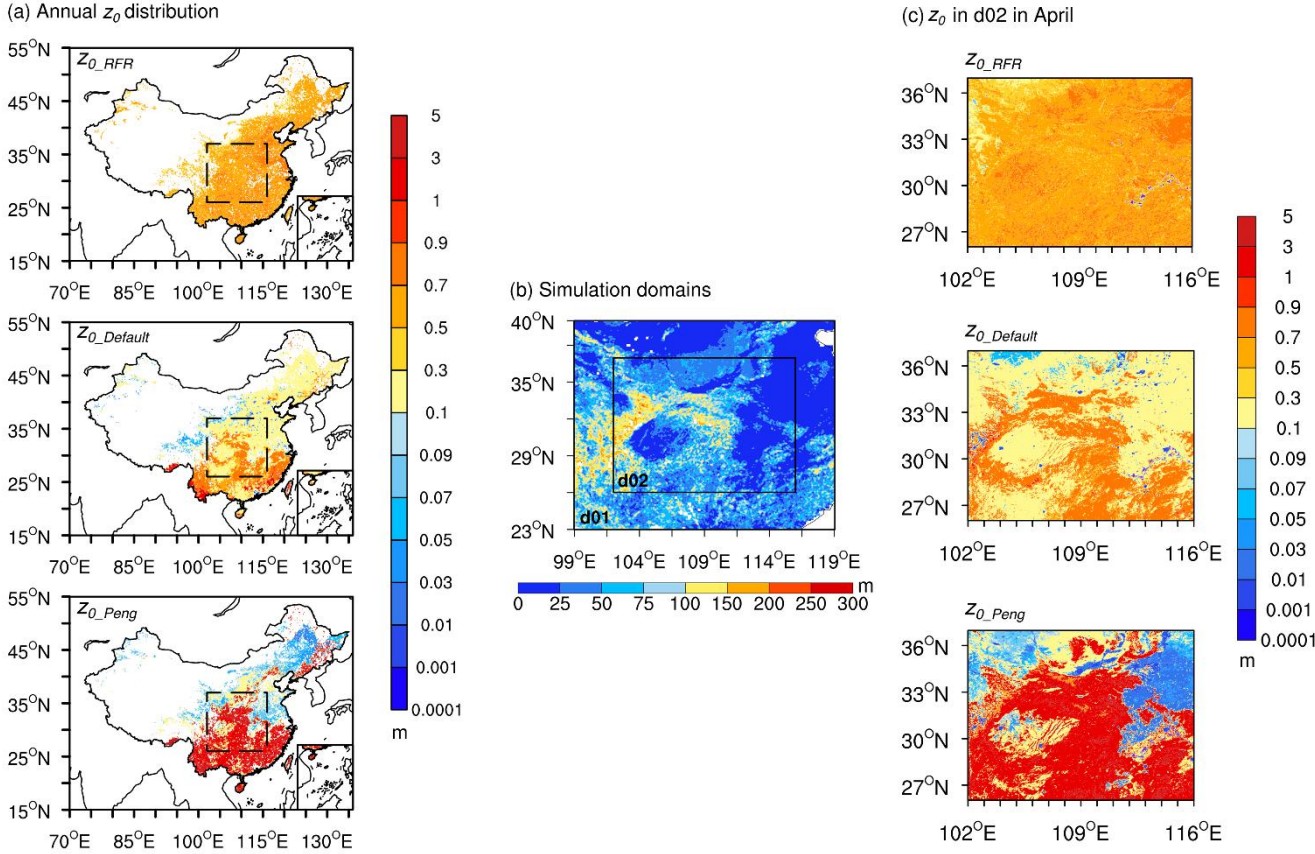

**Figure 4.** (a) Spatial distributions of annual mean $z_{0\_RFR}$, $z_{0\_Default}$, and $z_{0\_Peng}$. The dashed rectangular box indicates the simulation
domain (d02) in panel (b). (b) Nested simulation domains (d01: outer domain; d02: inner domain) with terrain standard deviation within
0.01° window (*TSD*) represented by color shading. (c) Spatial distributions of $z_0$ used in simulations over d02 in April.

### 3.3 Application of the produced $z_0$ datasets in wind speed simulation

To evaluate the performance of $z_{0\_RFR}$, we implemented it in the WRF model for wind speed simulations, as $z_0$ directly
affects near-surface wind speed. A 3-km simulation for April 2019 was conducted using the WRF model with $z_{0\_RFR}$ over
the regions outlined in Fig. 4a, which correspond to the d02 domain in Fig. 4b and represent the primary areas of $z_{0\_RFR}$
concentration. April was selected because it is the month with the highest average wind speed in the target domain (Fig. S3),
thus better reflecting the impact of $z_0$ on wind speed. For comparison, two additional simulations were performed: one
utilizing the WRF model's default roughness length ($z_{0\_Default}$) based on land cover types, and the other employing a recent
$z_0$ dataset ($z_{0\_Peng}$). In the northeastern, northern, and western regions of the d02 domain, both $z_{0\_Default}$ and $z_{0\_Peng}$ are
generally lower than $z_{0\_RFR}$ estimates, with $z_{0\_Peng}$ having even lower values than $z_{0\_Default}$ (Fig. 4c). However, this pattern
reverses in the southeastern areas and along the surrounding area of the Sichuan Basin, where both $z_{0\_Default}$ and $z_{0\_Peng}$
surpass $z_{0\_RFR}$ estimates, and notably, with $z_{0\_Peng}$ having significantly higher values than $z_{0\_Default}$ in these regions. These
discrepancies in $z_0$ would inevitably directly affect the accuracy of wind speed simulation. To evaluate the influence, we
conducted a comprehensive assessment on both 10-m and 100-m wind speed simulations, which represent typical heights for
meteorological observations and wind energy applications, respectively.

**3.3.1 Evaluation of the simulated 10-m wind speed**

We first compared the simulated 10-m wind speed with observations from 753 CMA stations in study areas (d02 domain),
showing that $z_{0\_RFR}$ significantly enhances the accuracy of simulations. The improvement due to $z_{0\_RFR}$ is evident in the
smaller *MBP* values of the simulated wind speed (Figs. 5a and S4) and the closer alignment of average wind speed with
observational data (Fig. 6a).
Specifically, the frequency histogram of *MBP* values reveals that the simulation results using $z_{0\_RFR}$ mostly fall within an
absolute *MBP* range of less than 30%, with a substantial proportion concentrated below 10%. In contrast, simulations
employing $z_{0\_Default}$ display a majority of *MBP* values exceeding 30%, while simulations using $z_{0\_Peng}$ are even poorer,
with a larger number of stations falling within higher *MBP* ranges (Fig. 5a). The improvement in 10-m wind speed induced
by $z_{0\_RFR}$ is primarily evident in relatively flat regions. As *TSD* increases, the improvement gradually diminishes (Fig. 5b).
$z_{0\_RFR}$ outperforms both $z_{0\_Default}$ and $z_{0\_Peng}$ when *TSD* does not exceed 50 m, while it shows superior performance to
$z_{0\_Default}$ and comparable results to $z_{0\_Peng}$ when *TSD* is greater than 50 m (Fig. 5c). Spatially, significant improvements are
observed in the relatively flat eastern and northern study areas, whereas limited enhancements are found in regions with
higher *TSD* surrounding the Sichuan Basin (Fig. S4). The limited improvement in relatively complex terrain arises because,
in addition to $z_0$, wind speed over these regions is influenced by multi-scale factors, including microscale terrain features
(Ge et al., 2025), turbulent orographic form drags (Beljaars et al., 2004; Jiménez and Dudhia, 2011; Zhou et al., 2018),
surface heating-induced mountain-valley circulations (Kim et al., 2021), mountain waves (Draxl, et al., 2021) and other
processes. Inaccurate parameterizations of these factors in numerical models can all lead to errors in wind speed simulations.
For the mean 10-m wind speed, simulations using $z_{0\_RFR}$ (2.26 m/s) show better agreement with the CMA observations (2.08
m/s), whereas simulations with $z_{0\_Default}$ and $z_{0\_Peng}$ show greater overestimations, producing mean wind speeds of 2.97
m/s and 2.90 m/s, respectively (Fig. 6a and Table 1). In other words, $z_{0\_RFR}$ decreases mean bias of 10-m wind speed by 79.8%
and 78.0% compared to $z_{0\_Default}$ and $z_{0\_Peng}$, respectively. Independent validations across 148 stations (Fig. 6b), from the
test subset in the generation of $z_{0\_RFR}$, further confirm the superiority of $z_{0\_RFR}$ (Fig. 6a). In addition, the improvements in
10-m wind speed were observed throughout the entire simulation period (Fig. 6c). Note that our experimental design,
employing a re-initialization strategy, means that 30 independent simulation experiments were conducted in April. Thus,
although the simulations were only conducted for a month, the consistent improvement across all days shows that the
enhancement achieved by $z_{0\_RFR}$ is robust. Moreover, the statistical metrics also show that the simulated 10-m wind speed
using $z_{0\_RFR}$ outperforms those using $z_{0\_Default}$ and $z_{0\_Peng}$ in temporal and spatial $MAB$ and $RMSE$ (Fig. 6d).

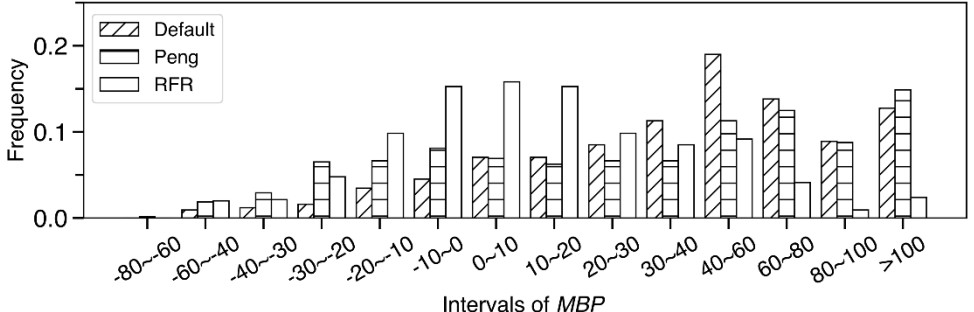

(a) Frequency of *MBP* intervals in simulated 10-m wind speed

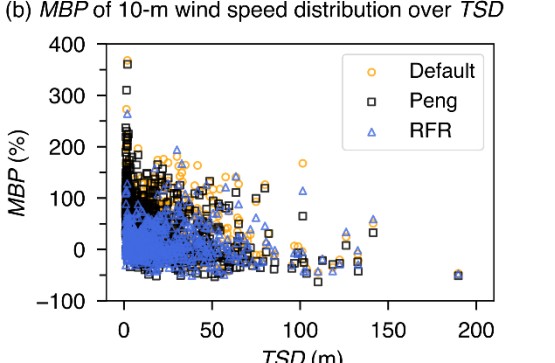

(b) *MBP* of 10-m wind speed distribution over *TSD*

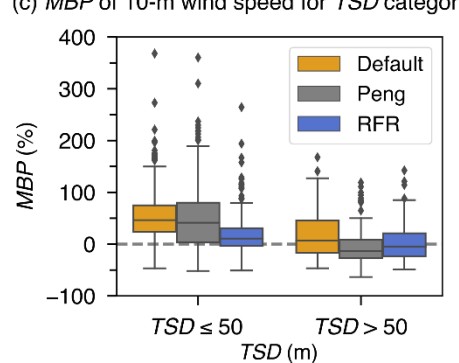

(c) *MBP* of 10-m wind speed for *TSD* categories


**Figure 5.** (a) Frequency distribution of *MBP* in simulated 10-m wind speed in April using $z_{0\_Default}$, $z_{0\_Peng}$, and $z_{0\_RFR}$ against
observations from CMA stations. (b) Distribution of *MBP* in 10-m wind speed as a function of *TSD*. (c) Box plot of *MBP* in 10-m wind
speed across different *TSD* bins.

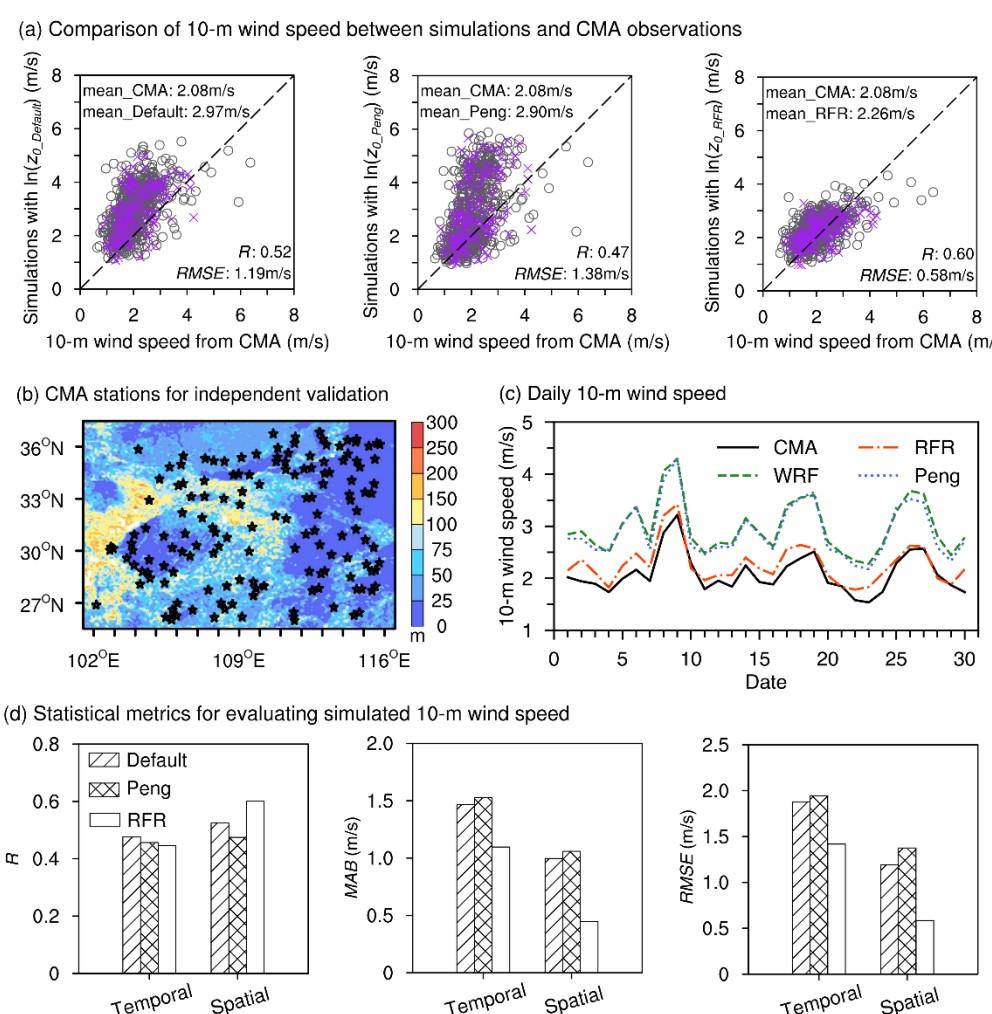

(a) Comparison of 10-m wind speed between simulations and CMA observations

(b) CMA stations for independent validation

(c) Daily 10-m wind speed

(d) Statistical metrics for evaluating simulated 10-m wind speed


**Figure 6.** (a) Comparisons of mean 10-m wind speed in April between the simulations using $z_{0\_Default}$, $z_{0\_Peng}$, and $z_{0\_RFR}$ versus observations from CMA stations. All points (grey circles and purple crosses) represent the 753 CMA stations within the d02 domain available for comparison, while the purple crosses represent the 148 stations utilized for independent validation, which were not used in training the $z_{0\_RFR}$ model. The corresponding wind speed means, $R$, and $RMSE$ of all stations are also indicated. (b) Distribution of the 148 independent CMA stations (black stars). Colored shaded areas represent $TSD$. (c) Comparison of daily mean 10-m wind speed between simulations and observations from 753 CMA stations. (d) Statistical metrics comparing simulated and observed 10-m wind speeds, including temporal and spatial $R$, $MAB$, and $RMSE$.

365

**Table 1.** Mean 10-m wind speed at 753 CMA stations and mean 100-m wind speed at 50 anemometer towers from simulations and observations. Simulations were performed using $z_{0\_Default}$, $z_{0\_Peng}$, and $z_{0\_RFR}$. Also shown is the percentage reduction in wind speed error ($PRE$) achieved by $z_{0\_RFR}$ relative to $z_{0\_Default}$ and $z_{0\_Peng}$.

| | $z_{0\_Default}$ | $z_{0\_Peng}$ | $z_{0\_RFR}$ | Observations |
|---|---|---|---|---|

| | | | | |
|---|---|---|---|---|
| Mean 10-m wind speed (m/s) | 2.97 | 2.90 | 2.26 | 2.08 |
| *PRE* in 10-m wind speed (%) | 79.8% | 78.0% | - | - |
| Mean 100-m wind speed (m/s) | 7.09 | 7.29 | 6.50 | 6.26 |
| *PRE* in 100-m wind speed (%) | 71.1% | 76.7% | - | - |

### 3.3.2 Evaluation of the simulated 100-m wind speeds

In addition to 10-m wind speed, the simulated 100-m wind speed was also improved through the use of $z_{0\_RFR}$ (Fig. 7a and Table 1). Compared to observations from 50 anemometer towers (Fig. 7b), with an average 100-m wind speed of 6.26 m/s, simulations based on $z_{0\_Default}$ and $z_{0\_Peng}$ overestimate the wind speed, with averages of 7.09 m/s and 7.29 m/s, respectively. However, the mean 100-m wind speed simulated using $z_{0\_RFR}$ is 6.50 m/s, closer to the observations (Table 1). This improvement using $z_{0\_RFR}$ reduces wind speed mean bias by 71.1% and 76.7% compared to $z_{0\_Default}$ and $z_{0\_Peng}$, respectively. Consistent with the performance of $z_{0\_RFR}$ at 10-m wind speed, the improvement in 100-m wind speed is more pronounced in relatively flat regions (Fig. 7c). The outliers in Fig. 7a, where wind speed biases remain significant despite using $z_{0\_RFR}$, are located in areas with higher *TSD*. Furthermore, similar to its performance at 10-m height, $z_{0\_RFR}$ demonstrates superior performance in simulated 100-m wind speed across both temporal and spatial metrics, with the exception of the temporal correlation coefficient (Fig. 7d). The relatively lower temporal $R$ is reasonable, as the improvement in wind speed induced by $z_0$ primarily stems from enhancements in the vertical profile.

In summary, the 30 independent simulation cases conducted for April demonstrate that the $z_0$ values derived from the combination of CMA observations and ERA5 data are highly reliable. The resulting gridded $z_0$ dataset significantly reduces uncertainties in mesoscale near-surface wind speed simulations, particularly over relatively flat high-roughness surface areas. To further validate the robustness of the $z_0$ estimation method and the resulting dataset, we conducted additional simulations for October 2019, a month characterized by generally weaker wind conditions (Fig. S3), using the same model configuration as in April. The results (Figs. S5-S7) also show consistent improvements when using $z_{0\_RFR}$. Station-wise correlations increase and errors decrease to a similar extent in both months, and the daily time series likewise show closer tracking of peaks and lulls. Taken together, these results further reinforce the reliability and applicability of the proposed $z_0$ estimation under varying meteorological conditions. They also indicate that although phenology-driven changes in canopy structure and seasonal circulation modulate wind speeds, the performance advantage of the proposed $z_0$ is not diminished.

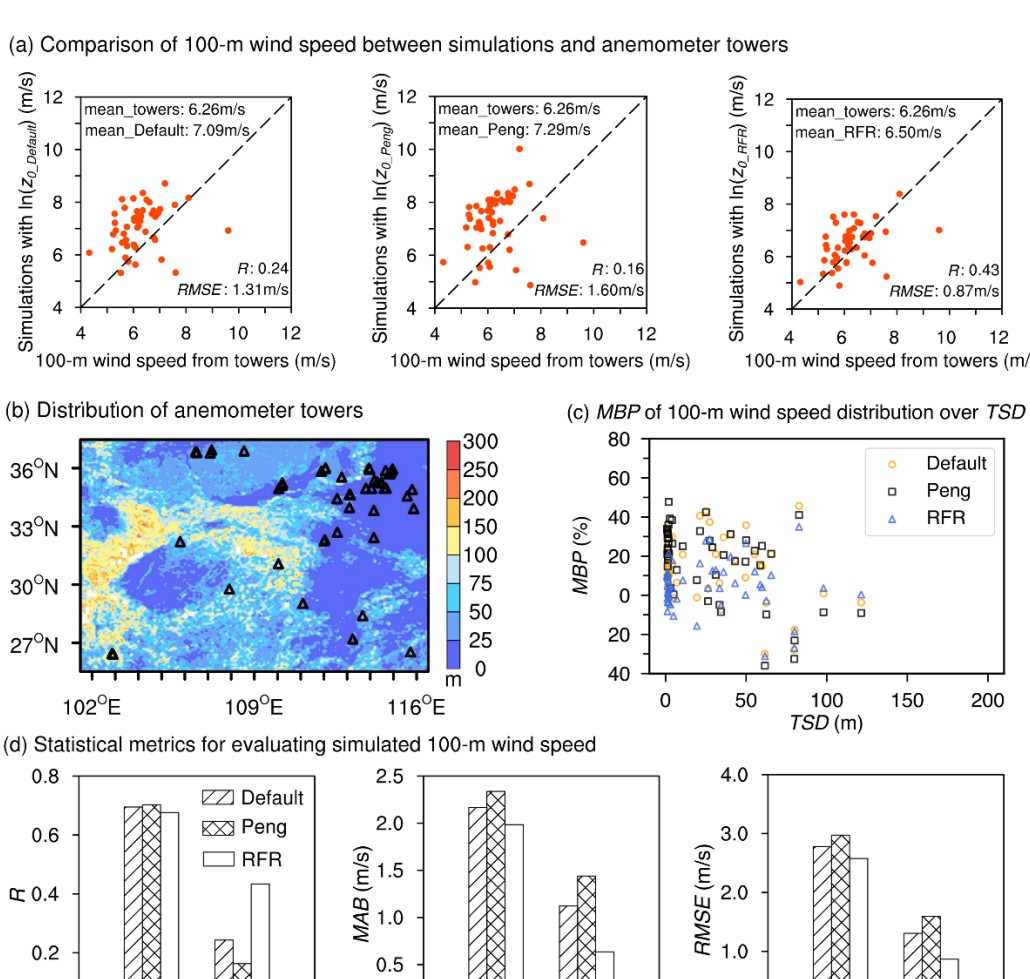

**Figure 7.** (a) Comparisons of mean 100-m wind speed in April between the simulations using $z_{0\_Default}$, $z_{0\_Peng}$, and $z_{0\_RFR}$ versus observations from anemometer towers. The corresponding wind speed means, $R$, and $RMSE$ of all towers are also indicated. (b) The locations of 50 anemometer towers (black triangles) utilized for 100-m wind speed evaluation. Colored shaded areas represent $TSD$. (c) Distribution of $MBP$ in 100-m wind speed as a function of $TSD$. (d) Statistical metrics comparing simulated and observed 100-m wind speeds, including temporal and spatial $R$, $MAB$, and $RMSE$.

## 4. Discussion

Here we discuss the sensitivity and generality of the site $z_0$ estimation approach with respect to the input simulation or reanalysis data, addressing concerns about potential methodological dependence on ERA5. Our study utilized ERA5 reanalysis data and CMA observations for initial $z_0$ estimation. Compared to traditional meteorological or morphological

methods, our approach can provide $z_0$ values at large spatial coverage and low cost, and these values lead to clear improvements in WRF-simulated wind speeds at both 10 m and 100 m above ground level. To assess whether the performance gain stems from improved $z_0$ representation rather than from alignment with ERA5 reanalysis data, we carried out two additional sets of evaluations.

First, we applied the same approach to estimate $z_0$ from WRF-simulated 10-m wind speed and the model's default $z_0$ values (0.03° × 0.03°), instead of ERA5. The $z_0$ values estimated using this alternative dataset were found to be highly similar to those derived from ERA5 (Fig. 8), indicating that the method is not inherently reliant on ERA5 as a data source. The primary advantage of using ERA5 lies in its extensive spatiotemporal coverage, which offers greater convenience and consistency with observational data. Meanwhile, although 10-m and 100-m winds over lands are not assimilated directly in ERA5, its 4D-Var system ingests a wide range of surface and upper-air observations that constrain boundary-layer structure and indirectly improve near-surface winds; this strengthens the credibility of using ERA5 as the reference field (Hersbach et al., 2020). However, the methodology itself is general and transferable to other datasets.

Moreover, the agreement between ERA5- and WRF-derived $z_0$ values suggests that the spatial extent represented by the estimated site-level $z_0$ values is not determined by the resolution of the reanalysis or simulation dataset used, but rather by the measurement height of wind observations at the stations. In this study, 10-m wind speeds from CMA stations were used. As a rule of thumb, the horizontal representativeness of wind measurements is approximately 10-100 times the measurement height. Therefore, $z_0$ values estimated from 10-m wind observations are reasonably representative at ~100 m-1 km scales, making the generation of 0.01° gridded $z_0$ datasets for use in mesoscale simulations both appropriate and justified, with no evident resolution dependence observed. We compared simulation results at different resolutions. Leveraging the nested modeling setup used in this study, the d01 domain with a 0.09° resolution was treated as the coarse-resolution simulation, while d02 at 0.03° served as the fine-resolution simulation. The results show that, even at the coarser resolution, our gridded $z_0$ dataset provides a clear advantage and substantially improves near-surface wind speed simulations (Fig. S8 and S9). However, for simulations at ~1 km resolution and finer, such as urban-scale wind modelling, our $z_0$ dataset cannot fully capture urban heterogeneity, because it did not incorporate key morphological parameters (e.g., building height and density) to distinguish between different urban forms. Therefore, an urban canopy model (UCM) would be a more appropriate choice. UCMs were conceived to operate at ~0.5-1 km grid spacing to bridge mesoscale forecasting (~$10^5$ m) with microscale transport/dispersion (~$10^0$ m) models (Tewari et al., 2006; Chen et al., 2010), and they have been widely applied and validated in subsequent urban studies (Lian et al., 2018; Salamanca et al., 2018; Wang et al., 2021). Therefore, our $z_0$ data are suitable and effective for mesoscale simulations at kilometer-level resolutions.

Second, we further validated the robustness of the refined $z_0$ dataset ($z_{0\_RFR}$) by conducting additional WRF simulations driven by the reanalysis from National Centers for Environmental Prediction (NCEP) instead of ERA5. These results (Fig. S10 and Table S1) still showed significant improvement in wind speed simulation performance when using $z_{0\_RFR}$,

consistent with those driven by ERA5. This cross-reanalysis consistency demonstrates that the benefits are attributable to the
improved surface representation through $z_{0\_RFR}$ refinement, not simply tuning to match ERA5-driven wind fields.
Taken together, these findings confirm that the $z_0$ estimation method proposed in this study is robust, flexible, and not
dependent on alignment with a specific reanalysis dataset. It provides a practical framework for $z_0$ estimation that can be
widely applied across different reanalysis/simulation datasets and observational data with consistent benefits. However, this
method is limited in regions with sparse or no surface weather stations. Notably, these regions, such as western and northern
China, are rich in wind resources and are key targets for wind energy development. Therefore, producing high-quality
gridded $z_0$ datasets in these regions warrants further study by exploring alternative data sources, such as anemometer tower
wind profiles, to supplement $z_0$ truth values (Wang et al., 2024).

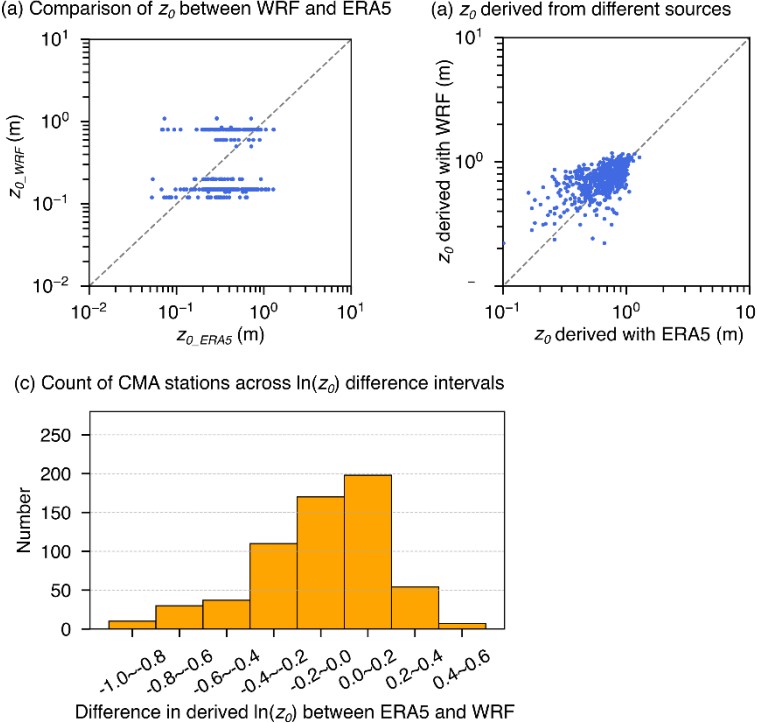


**Figure 8.** (a) Comparison of $z_0$ values from default WRF model ($z_{0\_WRF}$) and ERA5 ($z_{0\_ERA5}$). (b) Comparison of $z_0$ estimates using
different datasets. $z_0$ derived from WRF represents the estimated values based on WRF simulations (10-m wind speed and default $z_0$) and
CMA station observations (10-m wind speed) during April 2019, while $z_0$ derived from ERA5 denotes the estimates obtained in this study
using ERA5 reanalysis data in April. (c) Distribution of station counts across intervals of the difference in derived $\ln(z_0)$ ($\ln(z_0)$ derived
from ERA5 minus $\ln(z_0)$ derived from WRF).
The two assumptions used in the $z_0$ estimation are also discussed. Although these assumptions cannot be fully verified with
the available data, they are pragmatically motivated and indirectly supported by the improved performance of wind-speed
simulations using the resulting $z_0$ estimates. Assumption 1 posits that the near-surface wind-speed discrepancy between
ERA5 reanalysis and CMA observations is dominated by $z_0$ and that the influence of $z_0$ weakens with height, making ERA5
winds at higher levels within the surface layer comparable to observations. This is partly supported by the spatial pattern of
estimated $z_0$ (denser over eastern China, where 100-m wind-speed biases between ERA5 reanalysis and anemometer tower
observations are smaller (Figs. 1c and 2a)) and by a sensitivity test on the reference height (Figs. S11a and S11c). When re-
estimating annual-mean $z_{0\_CMA}$ at 150 m and 200 m, most stations show an absolute difference from the 100-m–based
estimate below 0.2, indicating broad consistency across heights. A minority of stations exhibit slightly larger deviation,
which may be influenced by local terrain complexity (Figs. S11b and S11d). Assumption 2 treats the effects of atmospheric
stability on wind speed as effectively similar in ERA5 and at CMA sites, allowing us to omit explicit stability corrections in
estimating $z_{0\_CMA}$. This simplification enhances methodological consistency and computational efficiency, and it is indirectly
supported by the validation of simulated winds. Moreover, prior work has shown that neutral log-law method can perform
comparably to stability-corrected scheme for vertical interpolation in U.S. wind-resource assessments (Duplyakin et al.,
2021), suggesting that such an approximate treatment seems feasible and a widely adopted simplification. Overall, although
neither assumption can be fully verified with the presently available data, their practical applicability is evidenced by
improved WRF wind-speed simulations. Future work, ideally leveraging multi-height wind profile observations and
coincident stability metrics could further test these assumptions, yield more precise $z_0$ estimates.
**5. Conclusion**
The representation of $z_0$ in numerical models, typically determined by land cover types, may lead to significant uncertainties
in wind speed simulations and predictions. Traditional methods for obtaining $z_0$ ground truth are mainly constrained by high
costs. In this study, we proposed a low-cost $z_0$ estimation method, allowing the acquisition of $z_0$ values at routine weather
stations.
Specifically, this approach leverages 10-m wind speed and $z_0$ values from ERA5 reanalysis data, along with observed 10-m
wind speeds at CMA stations, to derive optimal $z_0$ at stations by minimizing the difference in 100-m wind speeds between
reanalysis and observations. Here, the 100-m wind speed is expressed with 10-m wind speed and $z_0$ using similarity theory.
Based on this approach, we derived $z_0$ values at 1,837 CMA stations out of a total of 2,161 stations. These stations are
located in high-roughness surface regions, indicating the estimated $z_0$ values inherently include the effects of built-up and
tall vegetation.
To validate the reliability and practicality of the estimation method, we utilized a Random Forest Regression algorithm,
incorporating feature variables closely related to $z_0$, to develop a monthly gridded $z_0$ dataset for high-roughness surface
areas in China with a spatial resolution of $0.01° \times 0.01°$. The resulting $\ln(z_0)$ values mainly range from -1 to 0. Simulations
with WRF model show that, compared to the default $z_0$ in WRF and a recent gridded $z_0$ dataset developed by Peng et al.
(2022), the $z_0$ dataset constructed in this study has significantly improved the accuracy of near-surface wind speed
simulations in high-roughness surface areas, particularly in relatively flat regions. Evaluations against weather station data
and anemometer tower data show simulations with the new $z_0$ dataset mitigates mean bias of 10-m wind speed by 79.8% and
78.0%, and mean bias of 100-m wind speed by 71.1% and 76.7%, respectively, compared to the default $z_0$ in WRF and the
$z_0$ dataset from Peng et al. (2022).
In summary, this study developed a simple yet effective approach for correcting model $z_0$, addressing the limitations of
relying on empirical values assigned based on land cover types. The method shows particular effectiveness in $z_0$ correction
for high-roughness surface areas and offers valuable support for wind field-dependent studies and applications.

*Code and data availability.*
• Code required to conduct the analyses herein is available at https://doi.org/10.5281/zenodo.15108200 (Wang, 2025).

The datasets used in this study fall into two categories based on their accessibility:
1. Publicly Available Datasets (accessible via DOI/URL).
• The hourly wind speed data at 10 m and 100 m heights are obtained from the ERA5 reanalysis dataset (Hersbach et al.,
2020), accessible at https://doi.org/10.24381/cds.adbb2d47 (Hersbach et al., 2023b).
• For the gridded datasets of $z_0$ used in this study, $z_{0\_ERA5}$ (Hersbach et al., 2020) is available at
https://doi.org/10.24381/cds.f17050d7 (Hersbach et al., 2023a), while $z_{0\_Peng}$ (Peng et al., 2022) can be acquired by
contacting the corresponding authors.
• The initial and boundary conditions for the simulations are from the ERA5 dataset (Hersbach et al., 2020), which can
be downloaded from https://doi.org/10.24381/cds.adbb2d47 (Hersbach et al., 2023b) and
https://doi.org/10.24381/cds.bd0915c6 (Hersbach et al., 2023c).
• The digital elevation data, with a spatial resolution of 3 arc-seconds, are sourced from the Shuttle Radar Topography
Mission (SRTM) and can be downloaded from https://csidotinfo.wordpress.com/data/srtm-90m-digital-elevation-
database-v4-1/ (Jarvis et al., 2008).
• The urban-rural classification data (Li, X. et al., 2023) are available at https://doi.org/10.6084/m9.figshare.21716357.v6
(Li et al., 2022).
• The variance of the slope ($\overline{\theta^2}$) data can be obtained by contacting Zhou et al. (2018).
• The Leaf Area Index (LAI) data (Lin et al., 2023; Yuan et al., 2011) are accessible at
http://globalchange.bnu.edu.cn/research/laiv061 (Beijing Normal University Global Change Data Archive, 2022).
• The percent tree cover data (DiMiceli et al., 2022) can be obtained from https://doi.org/10.5067/MODIS/MOD44B.061
and https://search.earthdata.nasa.gov/search/granules?p=C2565805839-LPCLOUD&pg[0][v]=f&pg[0][gsk]=-
start_date&q=MOD44B&tl=1733462795.688!3!!&lat=-0.140625 (NASA EOSDIS, 2024a).
• The NDVI data (Didan, 2021) are available from https://doi.org/10.5067/MODIS/MOD13A3.061 and
https://search.earthdata.nasa.gov/search/granules?p=C2327962326-LPCLOUD&pg[0][v]=f&pg[0][gsk]=-
start_date&q=MOD13A3&tl=1732851935.718!3!!&lat=-0.140625 (NASA EOSDIS, 2024b).
• The NCEP forcing data (National Centers for Environmental Prediction/National Weather Service/NOAA/U.S.
Department of Commerce, 2025) are available from https://rda.ucar.edu/datasets/d083002/dataaccess/.
2. Restricted Datasets. We would like to clarify that the meteorological station data from the China Meteorological
Administration (CMA) and the anemometer tower data used in this study are not publicly accessible but can be accessed
through the following way. Specifically:
• The data from anemometer towers are provided by China State Shipbuilding Corporation Haizhuang Windpower Co.,
Ltd., however, they are not accessible publicly because of their commercial interests. These data can be obtained by
cooperation with the company.
• The hourly 10-m wind speed data at meteorological stations are from the China Meteorological Administration (CMA).
In accordance with the data policy of China, these data record are not directly accessible for public download via a
website. Nevertheless, individuals interested in obtaining detailed information about data acquisition can reach out to
the    China    Meteorological    Data    Service    Center    at    their    official    website
(http://data.cma.cn/en/?r=data/detail&dataCode=A.0012.0001, China meteorological data service centre, 2023).
*Author contributions.* All authors contributed to the study. JW and KY conceived the study and conducted the design; JW,
KY, and JL carried out data analyses; JW, XZ and XM performed the configuration of WRF model; WT processed data from
CMA stations; LY provided the data from anemometer towers; ZR conducted data collection and cleaning of anemometer
towers; JW and KY wrote the manuscript; all authors discussed, reviewed and edited the manuscript.
*Competing interests.* The contact author has declared that none of the authors has any competing interests.
*Disclaimer.* Publisher's note: Copernicus Publications remains neutral with regard to jurisdictional claims in published maps
and institutional affiliations.
*Financial support.* This work was supported by the National Natural Science Foundation of China (Grant Nos. 42475138
and 42361144875) and Huadian Xizang Energy Co., Ltd. (Grant No. 12IJD202400023).

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
