# Peer review of "Improvement of near-surface wind speed modeling through refined"

_EGUsphere, 2025_

## Author Comment (AC1)

Dear Dr. Shen,

We sincerely appreciate your time and effort for your comments on our manuscript, which may help us improve our work. Nevertheless, we would like to point out that three (Q1, Q2, and Q4) out of your four comments have already been extensively addressed in our original manuscript. Please see our point-by-point responses below.

Q1. The critical assumption that ERA5 100-m wind speed data closely aligns with observational data has not been sufficiently validated, especially for areas characterized by complex terrains or significant local environmental variations. The authors need to provide robust evidence supporting the applicability and limitations of this assumption.

**Response:** In our manuscript, we have addressed this concern by presenting a two-fold justification for our assumption. First, we evaluated ERA5 100-m winds with measurements from 589 wind towers across China, each providing months to years of data spanning different periods between 2004 and 2022. The results show that ERA5 exhibits a smaller mean bias percentage in eastern regions compared to western areas, supporting its higher reliability in eastern regions. This finding led us to focus primarily on weather stations in eastern China and to derive $z_0$ for 1,805 stations in these built-up regions. Please see this evaluation in Section 3.1. Second, we further validated our assumption through model experiments. The gridded $z_0$ dataset was tested in WRF simulations and independently evaluated against both unseen station data (10-m winds) and additional tower measurements (100-m winds). Both validation tests confirmed significant improvements in wind speed simulations. Please see the model improvement in Section 3.3.

Regarding the applicability of our assumption in complex terrain areas with significant local environmental variations, we have explicitly addressed this limitation in our study (Lines 281-285). Our analysis shows that the gridded $z_0$ produced based on station estimates provides only limited improvement for wind speed simulations in topographically complex regions. This suggests two possible explanations: (1) our fundamental assumption may not hold well in such areas, or more likely, (2) $z_0$ is not the sole determinant of wind speed in these regions. As discussed in our manuscript, wind patterns in complex terrain are governed by multi-scale physical processes including microscale terrain features, turbulent orographic form drag, thermally-driven mountain-valley circulations, and mountain wave dynamics. These processes may make the simple $z_0$-wind speed relationship invalid in flat terrain.

Q2. The observation dataset without homogenization from CMA has shown large bias in https://journals.ametsoc.org/view/journals/clim/36/11/JCLI-D-22-0445.1.xml. This may significantly affect the generalizability and accuracy of z0 estimations across broader geographic contexts. Direct usage of the CMA wind data would absolutely reduce the robustness of the study. Thus, the homogenization on near-surface wind data is necessary.

**Response:** We appreciate your reference to Zhang and Wang's study regarding wind speed inhomogeneity in CMA stations. Their work identified significant inhomogeneities with breakpoints concentrated in the late 1970s, mid-1990s, and early 2000s, but they do not affect our results. Our study exclusively uses CMA station data from 2015-2019, when the CMA network had already completed its transition to automated observations with standardized instruments. In addition, we conducted quality control procedures before use, including missing value screening, physical range

validation, and temporal consistency checks.

Q3. Although a Random Forest Regression model is employed, the sensitivity analysis of different feature variables lacks depth and clarity. The authors are encouraged to conduct comprehensive sensitivity analyses to clearly illustrate the theoretical rationale and practical implications of feature selection on model accuracy.

**Response:** In our study, we have conducted comprehensive sensitivity tests at every step of the random forest (RF) methodology to ensure the robustness of our results in Section 2.3 and Figure 3. Specifically, for data partitioning, we evaluated the impact of random seed selection when splitting the dataset into training and test subsets (Figure 3a); for parameter tuning, we systematically adjusted multiple key parameters (e.g., max_depth, n_estimators, min_samples_split, min_samples_leaf and so on) and provided detailed sensitivity analysis on the most influential parameter--the number of decision trees (Figure 3b); for model validation: a five-fold cross-validation approach was used to further verify the stability of our model (Figure 3c); for feature importance, we conducted thorough feature importance analysis to identify the dominant predictors (Figure 3e). These rigorous sensitivity tests confirm the reliability of our RF model. Please refer to Section 2.3 for a complete description of the methodology.

Q4. The validation of the model's performance is restricted to simulations for only one month, limiting the assessment of its robustness across different seasons or under varying long-term climatic conditions. The authors should include additional simulations covering multiple seasons or a full year to demonstrate the general applicability and reliability of their approach.

Given these substantial issues, I recommend rejecting this manuscript in its current form.

**Response:** We appreciate the reviewer's suggestion regarding the simulation period selection. Our choice to focus on April was motivated by both physical and practical considerations. As shown in Figure S3, April consistently exhibits the highest mean wind speeds across our study domain, making simulated wind speeds particularly sensitive to $z_0$ effects and thus ideal for evaluating our parameterization. To ensure robust results while managing computational constraints, we employed a carefully designed re-initialization approach where each 36-hour simulation (initialized daily at 12:00 LT (LT=UTC+8)) included a 12-hour spin-up period followed by 24 hours of analysis. This strategy produced 30 independent realizations, capturing diverse meteorological conditions throughout April. The consistent improvement in wind speed simulations across all cases (Section 3.3) strongly supports the reliability of our findings. While the current results are statistically robust, we may extend simulations to other months to further validate the general applicability of our $z_0$ dataset under varying climatic conditions.

We hope that we have addressed your concerns. We remain open to further feedback and are committed to improving the quality of our work.

Thank you very much!

Sincerely,
Jiamin Wang and Kun Yang
On behalf of all co-authors

---

## Author Comment (AC2)

**Responses to the Reviewer**

We would like to express our profound gratitude to you for your insightful comments and suggestions. Your expertise has significantly contributed to the enhancement of our study. In response to your valuable feedback, we have made corresponding revisions and additions to the manuscript. The detailed responses to each point raised are presented in the following sections. The responses are highlighted in blue, and the changes made in the manuscript are marked in red. We sincerely hope that these revisions adequately address your concerns.

**General Comments:**

This study estimated the aerodynamic roughness length ($z_0$) values using ERA5 analyses and weather station observations to improve the near-surface wind speed modeling. Technically, the Random Forest Regression algorithm is suitable for the estimation of $z_0$, and the results are encouraging, significantly improving the wind speed simulation in the WRF model. However, the evaluation of the improved $z_0$ on the WRF near-surface wind simulation was only for one month, and a longer time evaluation is needed. Therefore, I recommend Major Revision in this round.

**Response:** We are sincerely grateful for your positive feedback and constructive comments. Your comments have been thoroughly considered and have greatly contributed to the improvement of our manuscript. Our point-by-point responses are detailed below.

**Specific comments:**

**Major comments:**

1. The new estimated $z_0$ values were only evaluated for 1 month. A longer time evaluation should be conducted for a thorough evaluation.

**Response:** Thank you for your insightful comment. In our study, we initially evaluated the performance of the newly estimated aerodynamic roughness length ($z_0$) using wind simulations for the month of April in 2019. April was deliberately chosen as the primary evaluation period because it exhibits the highest mean wind speeds across our study domain (Fig. R1), making the simulated wind fields particularly sensitive to $z_0$ effects. This characteristic provides an ideal scenario for testing the impact and effectiveness of our proposed estimation method. To balance computational cost with scientific rigor, we implemented a re-initialization strategy whereby each 36-hour simulation was initialized daily at 12:00 LT (LT = UTC+8). Each simulation included a 12-hour spin-up period followed by 24 hours of analysis, yielding 30 independent realizations. This approach ensured the capture of a wide range of meteorological conditions while maintaining statistical independence among daily cases. As presented in Section 3.3,

the consistent improvement in simulated wind speeds across all April cases demonstrates the robustness of the newly estimated $z_0$.

To address the concern regarding longer-term evaluation, we additionally conducted WRF simulations for October 2019, a month characterized by generally weaker wind conditions (Fig. R1), using the same model configuration and evaluation framework as applied for April. The results from these additional simulations (Figs. R2-R4) further confirm the robustness of our method, as the use of the newly estimated $z_0$ values consistently improves the accuracy of simulated wind speeds.

[Figure]

**Figure R1** (**Figure S3 in the Supplement**). Monthly variations of the 10-m wind speed averaged over the d02 domain during 2015-2019 from ERA5.

The results of these additional simulations have been included in the Supplement (Figs. S5-S7), and a corresponding explanation has been incorporated into the revised manuscript in lines 332-338, replacing the original sentence: "In summary, the $z_0$ derived from the combination of CMA and ERA5 data shows high reliability, and the resulting gridded $z_0$ dataset in built-up areas can effectively reduce uncertainties in mesoscale near-surface wind speed simulations, especially over relatively flat built-up regions." with the following revised version: "In summary, the 30 independent simulation cases conducted for April demonstrate that the $z_0$ values derived from the combination of CMA observations and ERA5 data are highly reliable. The resulting gridded $z_0$ dataset significantly reduces uncertainties in mesoscale near-surface wind speed simulations, particularly over relatively flat built-up areas. To further validate the robustness of the $z_0$ estimation method and the resulting dataset, we conducted

additional simulations for October 2019, a month characterized by generally weaker wind conditions (Fig. S3), using the same model configuration as in April. The results (Figs. S5-S7) also show consistent improvements when using $z_{0\_RFR}$, further reinforcing the reliability and applicability of the proposed $z_0$ estimation approach under varying meteorological conditions."

[Figure]

**Figure R2 (Figure S5 in the Supplement)**. (a) Frequency distribution of *MBP* in simulated 10-m wind speed in October using $z_{0\_Default}$, $z_{0\_Peng}$, and $z_{0\_RFR}$ against observations from CMA stations. *MBP* was calculated as $[u_{simulations} - u_{CMA}]/u_{CMA} \times 100\%$. (b) Distribution of *MBP* in 10-m wind speed as a function of *TSD*. (c) Box plot of *MBP* in 10-m wind speed across different *TSD* bins.

[Figure]

**Figure R3 (Figure S6 in the Supplement).** (a) Comparisons of mean 10-m wind speed in October between the simulations using $z_{0\_Default}$, $z_{0\_Peng}$, and $z_{0\_RFR}$ versus observations from CMA stations. All points (grey circles and purple crosses) represent the 753 CMA stations within the d02 domain available for comparison, while the purple crosses represent the 155 stations utilized for independent validation, which were not used in training the $z_{0\_RFR}$ model. The corresponding wind speed means, correlation coefficients ($R$), and root mean square errors ($RMSE$) of all stations are also indicated. (b) Distribution of the 155 independent CMA stations (black stars). Colored shaded areas represent $TSD$. (c) Comparison of daily mean 10-m wind speed between simulations and observations from 753 CMA stations. (d) Statistical metrics comparing simulated and observed 10-m wind speeds, including temporal and spatial $R$, mean absolute bias ($MAB$, $\frac{1}{N}\sum_{i=1}^{N}\left|u_i^{simulation} - u_i^{observation}\right|$, where $N$ represents the number of hours for temporal $MAB$, and the number of stations for spatial $MAB$) and $RMSE$.

[Figure]

**Figure R4** (**Figure S7 in the Supplement**). (a) Comparisons of mean 100-m wind speed in October between the simulations using $z_{0\_Default}$, $z_{0\_Peng}$, and $z_{0\_RFR}$ versus observations from anemometer towers. The corresponding wind speed means, *R*, and *RMSE* of all towers are also indicated. (b) The locations of 48 anemometer towers (black triangles) utilized for 100-m wind speed evaluation. Colored shaded areas represent *TSD*. (c) Distribution of *MBP* in 100-m wind speed as a function of *TSD*. *MBP* was calculated as $[u_{simulations} - u_{towers}]/u_{towers} \times 100\%$. (d) Statistical metrics comparing simulated and observed 100-m wind speeds, including temporal and spatial *R*, *MAB*, and *RMSE*.

2. The grid-based $z_0$ statistics are only available in the inner domain. This indicates that the $z_0$ could only be improved where there are surface weather station observations. How to improve the $z_0$ destination in areas where there is no good coverage of surface weather station observations? More discussions should be included.

**Response:** We greatly appreciate your valuable question. We agree that the current implementation of our method is limited by the availability of surface weather station observations, which poses a challenge for estimating $z_0$ in areas with sparse or no such

coverage. Nevertheless, these under-observed regions, such as northern and northwestern China, are key zones for wind energy development. Thus, producing high-quality gridded $z_0$ datasets in these areas is not only of scientific interest but also crucial for enhancing the accuracy of wind speed simulations in practical applications.

A sufficient number of $z_0$ truth values is essential for generating such gridded datasets. The lack of $z_0$ truth values in station-sparse regions remains a major barrier. With the rapid growth of the wind energy industry, tens of thousands of such towers have been deployed for wind resource assessments. This development may offer a valuable opportunity to expand $z_0$ truth values and to construct a gridded $z_0$ dataset once these tower data are accessible.

We have included a discussion on this point in Section "4 Discussion" of the revised manuscript (lines 371-375), where we state: "However, this method is limited in regions with sparse or no surface weather stations. Notably, these regions, such as western and northern China, are rich in wind resources and are key targets for wind energy development. Therefore, producing high-quality gridded $z_0$ datasets in these regions warrants further study by exploring alternative data sources, such as anemometer tower wind profiles, to supplement $z_0$ truth values (Wang et al., 2024)."

**Minor comments:**

1. Line 39-41: It is a little bit causing here. Please revise it to be more clear.

**Response:** Thank you for your constructive reminder. We have revised "The utilization of wind energy in built-up areas also depends on wind speed distribution (Ishugah et al., 2014; Stathopoulos et al., 2018; Tasneem et al., 2020). Whether establishing wind farms in urban suburbs or integrating wind turbines into building designs, both can help to reduce generation load and the need for transmission infrastructure. Additionally, wind speed profoundly affects building design and urban planning (Hadavi and Pasdarshahri, 2020) and even the preservation of historical-cultural heritage (Li, Y. et al., 2023)." into "The utilization of wind energy in built-up areas also depends on wind speed distribution (Ishugah et al., 2014; Stathopoulos et al., 2018; Tasneem et al., 2020).

Proper utilization, through measures such as suburban wind farms or building-integrated turbines, can minimize the need for transmission infrastructure. Beyond energy considerations, wind speed characteristics play a critical role in urban design and planning, influencing both contemporary building practices (Hadavi and Pasdarshahri, 2020) and the preservation of historical-cultural heritage (Li, Y. et al., 2023)." in lines 37-42 of the revised manuscript.

2. Line 47: ERA5 is the analysis from a DA system. In my opinion, it is the blend of observations and model forecasts. Therefore, it is not proper to use it as an example.

**Response:** Thank you for your comment. We fully acknowledge that ERA5 is a reanalysis dataset generated through a data assimilation (DA) system, produced using 4D-Var DA and model forecasts in CY41R2 of the ECMWF Integrated Forecast System (IFS). However, it is important to note that the assimilated observations, especially over regions such as China, are relatively limited in spatial coverage. This partly explains the poor performance of ERA5 in representing wind speeds near the surface. Wang et al. (2024) evaluated the performance of ERA5 10-m wind speeds in China using data from both surface weather stations and anemometer towers, and found significant biases (Fig. R5). These biases indicate that the representation of near-surface wind conditions in ERA5 still heavily relies on the underlying model parameterizations, including the use of fixed $z_0$ based on land cover types. Therefore, we believe that using ERA5 as an example remains appropriate in the context of illustrating the limitations of $z_0$ treatment in current model frameworks.

[Figure]

**Figure R5 (Figure 4a from Wang et al. (2024))**. The distribution of $MBP$ of 10-m wind speed between ERA5 and measurements ( $(ERA5 - measurements)/measurements \times 100\%$ ). The dots and triangles represent the measurements from CMA stations and anemometer towers.

3. Line 54: What does it mean here "low-type" and "high-type"?

**Response:** Thank you for your insightful question. In this context, "low-type" and "high-type" vegetation refer to categories based on vegetation height. Specifically, "low-type" vegetation typically includes shorter land cover types such as grasslands and croplands, while "high-type" vegetation refers to taller vegetation such as forests. To enhance clarity, we have revised the sentence as follows: "In line with these findings, Luu et al. (2023) showed that the rise in $z_0$, caused by shifts from short vegetation to high vegetation and urbanization, partly contributes to the decline in mean and maximum surface wind speed over Western Europe." in lines 53-55 of the revised manuscript.

4. Line 87: Better to add surface weather station observations before CMA.

**Response:** Thank you for your useful suggestion. We have added "surface weather station observations" before "CMA" in lines 87-88 of the revised manuscript.

5. Line 192: This could be because of the altitude differences between observation sites and the model terrain.

**Response:** Thank you for your insightful comment. The altitude differences between observation sites and the model terrain could indeed contribute to the poor performance of ERA5 100-m wind speed data in these areas. To reflect this, we have revised the sentence as follows: "The exclusions of these stations can be attributed to the poor performance of ERA5 100-m wind speed data, which may result from altitude differences between the observation sites and the model terrain, thereby rendering our initial assumption, i.e. ERA5 100-m wind speed data are reliable for $z_0$ estimation, invalid in these areas." in lines 192-195 of the revised manuscript.

6. Line 227: What is the temporal coverage of this monthly $z_0$ dataset?

**Response:** Thank you for your insightful question. In this study, our primary objective was to propose a cost-effective method for estimating $z_0$ using weather station observations and reanalysis data. Accordingly, the monthly gridded $z_0$ dataset we produced, referred to as $z_{0\_RFR}$, was mainly intended to demonstrate the feasibility and effectiveness of the $z_0$ estimation approach through wind speed simulations. For this purpose, the $z_{0\_RFR}$ dataset was generated for the year 2019 as a representative example. It is important to note, however, that the Random Forest Regression (RFR) model developed for generating the gridded $z_{0\_RFR}$ dataset is not limited to a specific year. It can readily be applied to other years, provided that the corresponding input features are available.

To clarify this point, we have revised the manuscript and added the following statements: "As a representative example, the $z_{0\_RFR}$ dataset was generated for the year 2019, and its spatial coverage is shown in Fig. 2d." (lines 231-232) and "Although 2019 was chosen for demonstration, the RFR model itself is year-independent and can be applied to other years, provided that the required input features are available." (lines 235-237).

7. Figure 5: better to a reference line of y = 0 in panel (c) for reference, indicating which has a smaller bias.

**Response:** Thank you for your constructive suggestion. We have added a reference line at $y = 0$ in Fig. 5c to indicate the direction and magnitude of the bias more clearly.

8. Line 317: The values are significantly large when verified against the Mean values. However, if you take a deep look at Fig. 7d, the improvements are not that large from the perspective of MAB and RMSE.

**Response:** Thank you for your insightful comment. In line 317 of the original manuscript, we stated: "This improvement using $z_{0\_RFR}$ reduces wind speed mean bias by 85.7% and 88.1% compared to $z_{0\_Default}$ and $z_{0\_Peng}$, respectively.", which indeed shows a substantial improvement. While this appears to contrast with the results presented in Fig. 7d, where the reductions in *MAB* and *RMSE* seem less pronounced,

this discrepancy arises from the use of different evaluation metrics. Specifically, the percentage reduction of wind speed mean bias refers to the relative decrease in mean error. For example, in the simulation based on $z_{0\_Default}$ the average 100-m wind speed is 7.10 m/s; while using $z_{0\_RFR}$, it is 6.38 m/s. The corresponding observed value from the anemometer towers is 6.26 m/s. Thus, the mean bias is reduced from 0.84 m/s $(7.10 - 6.26)$ to 0.12 m/s $(6.38 - 6.26)$, leading to a bias reduction of $(0.84 - 0.12) \div 0.84 \times 100\% = 85.7\%$.

In addition, even when evaluated using the *MAB* and *RMSE* metrics shown in Fig. 7d, the improvements brought by $z_{0\_RFR}$ are still considerable. Specifically, in the spatial dimension, the 100-m wind speed simulations based on $z_{0\_Default}$ and $z_{0\_Peng}$ show *MAB* values of 1.12 m/s and 1.47 m/s, respectively, while the simulation using $z_{0\_RFR}$ yields a significantly lower *MAB* of 0.58 m/s. Similarly, the corresponding *RMSE* values are 1.31 m/s for $z_{0\_Default}$, 1.63 m/s for $z_{0\_Peng}$, and 0.82 m/s for $z_{0\_RFR}$. Although the improvements in the temporal dimension are not as pronounced as those in the spatial dimension, they are still evident. These results further confirm the overall improvement achieved by incorporating the $z_{0\_RFR}$.

To enhance clarity, we have added the formula used to calculate the percentage reduction in wind speed mean bias to the revised manuscript, as shown in the caption of Table 1: "The percentage reduction in wind speed error is caused by $z_{0\_RFR}$, compared to $z_{0\_Default}$ and $z_{0\_Peng}$, which is calculated as $\left[\left|\bar{u}_{z_{0\_*}} - \bar{u}_{observation}\right| - \left|\bar{u}_{z_{0\_RFR}} - \bar{u}_{observation}\right|\right] / \left|\bar{u}_{z_{0\_*}} - \bar{u}_{observation}\right| \times 100\%$, where $\bar{u}_{z_{0\_*}}$ represents $\bar{u}_{z_{0\_Default}}$ or $\bar{u}_{z_{0\_Peng}}$, and $\bar{u}$ denotes the mean 10-m or 100-m wind speed from simulations based on $z_{0\_Default}$, $z_{0\_Peng}$, and $z_{0\_RFR}$, as well as from observations (CMA stations or anemometer towers)."

9. Figure 7: Better to add statists of mean/rms/r in the panels of (a). For (d), the units of MAB is not m/s, likely %.

**Response:** Thank you for your suggestion. We have added the statistics of mean, *RMSE,* and correlation coefficient (*R*) to both Fig. 6a and Fig. 7a in the revised manuscript.

*MAB* refers to mean absolute bias, which is calculated as $\frac{1}{N}\sum_{i=1}^{N}\left|u_i^{simulation} - u_i^{observation}\right|$, where $N$ represents the number of stations for spatial *MAB*, and the number of hours for temporal *MAB*. Therefore, the unit of *MAB* is m/s.

To enhance clarity, we have included the formula for *MAB* at its first occurrence, in the caption of Fig. 6 in the revised manuscript, as follows: "Figure 6. (a) Comparisons of mean 10-m wind speed in April between the simulations using $z_{0\_Default}$, $z_{0\_Peng}$, and $z_{0\_RFR}$ versus observations from CMA stations. All points (grey circles and purple crosses) represent the 753 CMA stations within the d02 domain available for comparison, while the purple crosses represent the 155 stations utilized for independent validation, which were not used in training the $z_{0\_RFR}$ model. The corresponding wind speed means, correlation coefficients ($R$), and root mean square errors (*RMSE*) of all stations are also indicated. (b) Distribution of the 155 independent CMA stations (black stars). Colored shaded areas represent *TSD*. (c) Comparison of daily mean 10-m wind speed between simulations and observations from 753 CMA stations. (d) Statistical metrics comparing simulated and observed 10-m wind speeds, including temporal and spatial $R$, mean absolute bias (*MAB*, $\frac{1}{N}\sum_{i=1}^{N}\left|u_i^{simulation} - u_i^{observation}\right|$, where $N$ represents the number of hours for temporal *MAB*, and the number of stations for spatial *MAB*) and *RMSE*."

**Reference**

Wang, J., Yang, K., Yuan, L., Liu, J., Peng, Z., Ren, Z. and Zhou, X.: Deducing aerodynamic roughness length from abundant anemometer tower data to inform wind resource modeling, Geophys. Res. Lett., 51, e2024GL111056, doi:10.1029/2024GL111056, 2024.

---

## Author Comment (AC3)

**Responses to the Reviewer**

We would like to express our profound gratitude to you for your insightful comments and suggestions. Your expertise has significantly contributed to the enhancement of our study. In response to your valuable feedback, we have made corresponding revisions and additions to the manuscript. The detailed responses to each point raised are presented in the following sections. The responses are highlighted in blue, and the changes made in the manuscript are marked in red. We sincerely hope that these revisions adequately address your concerns.

**General Comment:**

This manuscript presents a novel and practical approach to improving the simulation of near-surface wind speed over built-up areas by refining the aerodynamic roughness length ($z_0$) using a combination of ERA5 reanalysis and ground-based observations from the China Meteorological Administration (CMA). The authors developed a high-resolution monthly gridded $z_0$ dataset by applying a Random Forest Regression algorithm, and demonstrated its effectiveness through WRF simulations. The study is timely and potentially impactful for urban climate modeling and wind-related applications.

While the manuscript introduces a potentially useful methodology, the current version does not provide sufficient critical evaluation or methodological transparency. To be suitable for publication, the manuscript requires revision, including clarification of the observational setup, deeper theoretical consideration of the methodology's assumptions, and further analyses related to model resolution and $z_0$ scale dependency.

**Response:** We would like to express our sincere gratitude for your positive feedback and insightful comments and suggestions. These have significantly enhanced the quality of our manuscript. We have carefully considered all your points. In the following sections, we provide a detailed response to each of your comments.

**Major comments:**

1. Uncertainty about CMA Wind Observation Heights: The manuscript assumes that CMA stations provide 10-m wind speed observations. However, there is no clear documentation or justification of this assumption in the text. Are all CMA anemometers calibrated and installed precisely at 10 m above ground level? Given that the accuracy of $z_0$ estimation strongly depends on the reference height of the wind speed, this should be clarified and supported by official metadata or references. Otherwise, the credibility of the derived $z_0$ values may be significantly undermined.

**Response:** Thank you for your question. All CMA wind speed observations used in this study were indeed measured at the standard height of 10 meters above ground level,

as officially specified in the "China Surface Climate Data Hourly Value Dataset" provided by the China Meteorological Administration (Table R1). In addition, the $z_0$ estimated from these stations have been independently validated using wind speed simulations against both other CMA stations and anemometer tower observations. The validation results demonstrate that the derived $z_0$ values lead to significant improvements in simulated wind speeds, thereby supporting the overall reliability of our $z_0$ estimates.

**Table R1**. Selected fields from the China Surface Climate Data Hourly Value Dataset provided by the China Meteorological Administration (CMA).

| No. | Name | Data Type | Field Name | Unit |
|-----|------|-----------|------------|------|
| 1 | Station ID | Number(5) | V01000 | — |
| 5 | Year | Number(4) | V04001 | — |
| 6 | Month | Number(2) | V04002 | — |
| 7 | Day | Number(2) | V04003 | — |
| 8 | Hour | Number(2) | V04004 | — |
| 9 | Station Pressure | Number(6) | V10004 | 0.1 hPa |
| 11 | Air Temperature | Number(6) | V12001 | 0.1 °C |
| 19 | Precipitation | Number(6) | V13011 | 0.1 mm |
| 21 | Wind Direction (at 10 m above ground) | Number(6) | V11011 | 16 directions |
| 22 | Wind Speed (at 10 m above ground) | Number(6) | V11012 | 0.1 m/s |

2. Circular Logic in Using ERA5 to Derive $z_0$ and Then Evaluating WRF Performance: The method uses ERA5 as the basis to derive optimal $z_0$ values, and then uses these $z_0$ values in WRF to simulate wind fields, which are subsequently compared to CMA observations. However, since the $z_0$ is essentially tuned to ERA5 wind characteristics, and WRF is driven by ERA5 data, it is not surprising that the WRF simulations become closer to observations. This circular logic reduces the strength of the validation. A deeper discussion is needed in the Discussion section to acknowledge this methodological dependency and to better clarify to what extent the improvements stem from $z_0$ refinement as opposed to alignment with the reanalysis base.

**Response:** Thank you for raising this important point. We address the concern about potential circular logic from three perspectives, to demonstrate that the improvement on wind speed primarily stems from the refinement of $z_0$, rather than simply from alignment with the reanalysis dataset.

First, the $z_0$ values were estimated at 1,805 CMA station locations using CMA-observed 10-m wind speeds, ERA5 10-m wind speeds, and ERA5 $z_0$. Based on these $z_0$ estimates, we used 80% of the data to train a machine learning model and construct a gridded $z_0$ dataset, while the remaining 20% were reserved for independent validation. This gridded dataset (denoted as $z_{0\_RFR}$) was then used in WRF simulations. In evaluating the WRF results, we considered wind speeds at both 10 m and 100 m, which are representative of meteorological observations and wind energy applications, respectively. At 10 m, simulation performance was assessed at all 753 CMA stations in the domain, including both the 598 training stations and the 155 independent validation stations. The results show that WRF simulations using $z_{0\_RFR}$ outperform those using the default WRF dataset ($z_{0\_Default}$) and a latest dataset ($z_{0\_Peng}$), as demonstrated in Fig. 6 of the manuscript. At 100 m, further validation was performed using wind measurements from anemometer towers, which were completely independent from both the CMA stations used in training the $z_0$ model and the $z_0$ estimation process. These results (Fig. 7 of the manuscript) also confirm the superiority of $z_{0\_RFR}$, strengthening the claim that the improvements stem from the enhanced representation of $z_0$ rather than any alignment with ERA5 data.

Second, according to your suggestion, we conducted an additional WRF simulation using NCEP reanalysis data instead of ERA5 as the driving input, while keeping all other model settings identical. The results (Figure R1 and Table R2) are consistent with those obtained using ERA5 forcing data (Figure 6a and 7a, and Table 1 in the manuscript), indicating that $z_{0\_RFR}$ improves wind speed simulations. This strongly suggests that the improvements are not a result of alignment between the tuned $z_0$ values and the ERA5 data, but rather due to the intrinsic quality of the refined $z_0$ dataset itself.

[Figure]

**Figure R1** (**Figure S8 in the Supplement**). Comparison between simulated wind speeds and observations, with WRF driven by NCEP reanalysis data. (a) Comparisons of mean 10-m wind speed in April between the simulations using $z_{0\_Default}$, $z_{0\_Peng}$, and $z_{0\_RFR}$ versus observations from CMA stations. All points (grey circles and purple crosses) represent the 753 CMA stations within the d02 domain available for comparison, while the purple crosses represent the 155 stations utilized for independent validation, which were not used in training the $z_{0\_RFR}$ model. The corresponding wind speed means, correlation coefficients (*R*), and root mean square errors (*RMSE*) of all stations are indicated. (b) Comparisons of mean 100-m wind speed in April between the simulations using $z_{0\_Default}$, $z_{0\_Peng}$, and $z_{0\_RFR}$ versus observations from anemometer towers. The corresponding wind speed means, *R*, and *RMSE* of all towers are also indicated.

**Table R2** (**Table S1 in the Supplement**). The mean 10-m wind speed from simulations and observations at 753 CMA stations, and the mean 100-m wind speed from simulations and observations at 50 anemometer towers. The simulations were conducted using $z_{0\_Default}$, $z_{0\_Peng}$, and $z_{0\_RFR}$, respectively, with NCEP reanalysis data used as the driving input for the WRF model. The percentage reduction in wind speed error is caused by $z_{0\_RFR}$, compared to $z_{0\_Default}$ and $z_{0\_Peng}$, which is calculated as $\left[\left|\bar{u}_{z_{0\_*}}-\bar{u}_{observation}\right|-\left|\bar{u}_{z_{0\_RFR}}-\bar{u}_{observation}\right|\right]/\left|\bar{u}_{z_{0\_*}}-\bar{u}_{observation}\right|\times100\%$, where $\bar{u}_{z_{0\_*}}$ represents

$\bar{u}_{z_{0\_Default}}$ or $\bar{u}_{z_{0\_Peng}}$, and $\bar{u}$ denotes the mean 10-m or 100-m wind speed from simulations based on $z_{0\_Default}$, $z_{0\_Peng}$, and $z_{0\_RFR}$, as well as from observations (CMA stations or anemometer towers).

| | $z_{0\_Default}$ | $z_{0\_Peng}$ | $z_{0\_RFR}$ | Observations |
|---|---|---|---|---|
| Mean 10-m wind speed (m/s) | 2.94 | 2.86 | 2.14 | 2.08 |
| Percentage reduction in 10-m wind speed error caused by $z_{0\_RFR}$ (%) | 93.0% | 92.3% | - | - |
| Mean 100-m wind speed (m/s) | 6.89 | 7.10 | 6.21 | 6.26 |
| Percentage reduction in 100-m wind speed error caused by $z_{0\_RFR}$ (%) | 92.1% | 94.0% | - | - |

Third, we have examined whether the effectiveness of the proposed $z_0$ estimation method is inherently dependent on the use of ERA5 data in Section "4 Discussion" of the manuscript. We applied the same approach to estimate $z_0$ with 10-m wind speed and default $z_0$ values from the WRF model itself, instead of ERA5. The estimated $z_0$ values based on this alternative dataset are similar to those derived from ERA5 (Figure 8b in the manuscript). This demonstrates that the validity of our $z_0$ estimation method does not rely on alignment with any specific reanalysis dataset, but rather reflects the robustness and general applicability of the method itself.

In summary, through independent validation at both 10 m and 100 m heights, additional experiments using alternative reanalysis inputs (NCEP instead of ERA5), and further tests employing non-ERA5-based inputs for $z_0$ estimation, we consistently demonstrate that the improved WRF performance arises from the refined characterization of $z_0$ itself. These results collectively confirm that the effectiveness of our method is not due to any circular logic or alignment with a specific reanalysis dataset, but rather reflects the intrinsic value and robustness of the proposed $z_0$ refinement approach. Accordingly, we have reorganized Section "4 Discussion" in the revised manuscript.

It was originally: "Here we discuss the sensitivity of the site $z_0$ estimates to the used simulation/reanalysis data. Our study utilized ERA5 reanalysis and CMA observations for $z_0$ estimation. Compared to traditional meteorological and morphological methods, the approach can obtain $z_0$ values at most locations at a low cost, and these values demonstrate satisfactory performance in wind speed simulation. Here we show that the

method is not restricted to using ERA5 reanalysis data. When it is applied to 10-m wind speed and default $z_0$ from WRF model, we can estimate $z_0$ similarly. The resulting $z_0$ estimates are comparable to those based on ERA5 (Fig. 8). The primary advantage of ERA5 is its extensive spatiotemporal coverage, which facilitates better alignment with observational data. In contrast, obtaining WRF simulation data with the same spatiotemporal coverage would require considerable computational resources. Therefore, the proposed method in this paper is a robust $z_0$ estimation approach that can be widely applied to different reanalysis datasets and observational data, offering high flexibility and practicality for aerodynamic roughness length estimation."

It is now revised to: "Here we discuss the sensitivity and generality of the site $z_0$ estimation approach with respect to the input simulation or reanalysis data, addressing concerns about potential methodological dependence on ERA5. Our study utilized ERA5 reanalysis data and CMA observations for initial $z_0$ estimation. Compared to traditional meteorological or morphological methods, our approach can provide $z_0$ values at large spatial coverage and low cost, and these values lead to clear improvements in WRF-simulated wind speeds at both 10 m and 100 m above ground level. To assess whether the performance gain stems from improved $z_0$ representation rather than from alignment with ERA5 reanalysis data, we carried out two additional sets of evaluations.

First, we applied the same approach to estimate $z_0$ from WRF-simulated 10-m wind speed and the model's default $z_0$ values (0.03° × 0.03°), instead of ERA5. The $z_0$ values estimated using this alternative dataset were found to be highly similar to those derived from ERA5 (Fig. 8), indicating that the method is not inherently reliant on ERA5 as a data source. The primary advantage of using ERA5 lies in its extensive spatiotemporal coverage, which offers greater convenience and consistency with observational data; however, the methodology itself is general and transferable to other datasets. Moreover, the agreement between ERA5- and WRF-derived $z_0$ values suggests that the spatial representativeness of the estimated site-level $z_0$ values is not determined by the resolution of the reanalysis or simulation dataset used, but rather by the measurement height of wind observations at the stations. In this study, 10-m wind

speeds from CMA stations were used. As a rule of thumb, the horizontal representativeness of wind measurements is approximately 100 times the measurement height. Therefore, $z_0$ values estimated from 10-m wind observations are reasonably representative at ~1 km scales, making the generation of 0.01° gridded $z_0$ datasets for use in mesoscale simulations both appropriate and justified.

Second, we further validated the robustness of the refined $z_0$ dataset ($z_{0\_RFR}$) by conducting additional WRF simulations driven by the reanalysis from National Centers for Environmental Prediction (NCEP) instead of ERA5. These results (Fig. S8 and Table S1) still showed significant improvement in wind speed simulation performance when using $z_{0\_RFR}$, consistent with those driven by ERA5. This cross-reanalysis consistency demonstrates that the benefits are attributable to the improved surface representation through $z_{0\_RFR}$ refinement, not simply tuning to match ERA5-driven wind fields.

Taken together, these findings confirm that the $z_0$ estimation method proposed in this study is robust, flexible, and not dependent on alignment with a specific reanalysis dataset. It provides a practical framework for $z_0$ estimation that can be widely applied across different reanalysis/simulation datasets and observational data with consistent benefits. However, this method is limited in regions with sparse or no surface weather stations. Notably, these regions, such as western and northern China, are rich in wind resources and are key targets for wind energy development. Therefore, producing high-quality gridded $z_0$ datasets in these regions warrants further study by exploring alternative data sources, such as anemometer tower wind profiles, to supplement $z_0$ truth values (Wang et al., 2024)."

3. Lack of Resolution-Dependent $z_0$ Consideration: The aerodynamic roughness length is known to be resolution-dependent due to varying representations of land cover and orography. However, the manuscript does not address why a single $z_0$ value (derived from coarser ERA5 resolution) is applied across finer-resolution WRF simulations. A justification is needed as to why scale-dependent roughness parameters were not considered, especially when moving from ERA5 (~30 km) to WRF (3 km).

Moreover, higher-resolution simulations are expected to better resolve local features influencing $z_0$. Has the relationship between horizontal resolution and $z_0$ been explored in this study? Such an analysis would greatly strengthen the work, and I recommend adding or expanding this aspect if possible.

**Response:** Thank you for your valuable question. In this study, we proposed a low-cost $z_0$ estimation method, allowing the acquisition of $z_0$ values at routine weather stations. Specifically, this approach leverages 10-m wind speed and $z_0$ values from ERA5 reanalysis data, along with observed 10-m wind speeds at CMA stations, to derive optimal $z_0$ at stations by minimizing the difference in 100-m wind speeds between reanalysis and observations. Here, the 100-m wind speed is expressed with 10-m wind speed and $z_0$ using similarity theory.

Regarding the use of ERA5 data in the estimation, we would like to clarify that although we introduced the assumption that the 100-m wind speed from ERA5 is comparable to that from observations, 100-m wind speed was not directly used in the actual estimation process of $z_0$. Rather, this assumption served to conceptually support the feasibility of using ERA5 10-m wind speed and $z_0$ information to estimate $z_0$ values at observational sites. This assumption implies that the influence of $z_0$ on wind speed at 100 m is relatively small. While similar assumptions could be made using reanalysis datasets providing wind speeds at even higher levels (e.g., 200 m), we chose to use the 100-m level because ERA5 provides wind speed at this height and there are anemometer tower data at 100 m available for preliminary validation of this assumption. Therefore, this assumption is not constrained by the spatial resolution of the dataset used. In practice, our method estimates $z_0$ using $0.25° \times 0.25°$ gridded 10-m wind speed and $z_0$ data from ERA5. Essentially, what we utilize is the relationship between the wind profile and $z_0$ as represented in ERA5 through similarity theory. The horizontal resolution of ERA5 does not affect the estimated $z_0$ values at individual stations. To demonstrate this, we substituted ERA5 with higher-resolution WRF outputs $(0.03° \times 0.03°)$ to re-estimate $z_0$, and the results remained consistent, as discussed in Section "4 Discussion" of the manuscript.

More importantly, the spatial representativeness of the derived $z_0$ values is determined primarily by the measurement height of wind observations, rather than the resolution of the background dataset. As a rule of thumb, the effective fetch area influencing a wind measurement is approximately 100 times the measurement height. Since we used 10-m wind speed data from CMA stations, the estimated $z_0$ values are representative of a footprint of ~1 km. Therefore, applying these $z_0$ values to kilometer-scale simulations is scale-consistent and appropriate. In addition, we have previously emphasized in the manuscript that the $z_0$ values derived in this study are intended for use in mesoscale simulations (see lines 81-83 ("This study contributes to the advancement of mesoscale wind speed simulation over built-up environments, which can promote wind field-dependent studies, such as urban planning, wind energy utilization, and air quality management.") and 333-334 ("The resulting gridded $z_0$ dataset significantly reduces uncertainties in mesoscale near-surface wind speed simulations, particularly over relatively flat built-up areas.).

Based on the above, the updated Discussion section (lines 357-363) now further elaborates on this point: "Moreover, the agreement between ERA5- and WRF-derived $z_0$ values suggests that the spatial representativeness of the estimated site-level $z_0$ values is not determined by the resolution of the reanalysis or simulation dataset used, but rather by the measurement height of wind observations at the stations. In this study, 10-m wind speeds from CMA stations were used. As a rule of thumb, the horizontal representativeness of wind measurements is approximately 100 times the measurement height. Therefore, $z_0$ values estimated from 10-m wind observations are reasonably representative at ~1 km scales, making the generation of 0.01° gridded $z_0$ datasets for use in mesoscale simulations both appropriate and justified."

---

## Author Comment (AC4)

**Clarification on corrections to EGUSPHERE-2025-1513**

Dear Editor and Reviewers,

During a post-submission workflow check, we identified a time misalignment in the processing of meteorological station observations used for aerodynamic roughness length  $(z_0)$  estimation and model evaluation. This issue has now been corrected. After reprocessing the data, we found that the correction causes minor numerical differences in several evaluation metrics and figures, but the conclusions remain unchanged.

**Summary of changes:**

- 1. Corrected the time alignment of CMA station observations.
- 2. Recomputed the  $z_0$  estimates,  $z_0$  gridded dataset and model evaluation metrics.
- 3. Updated several figures and tables and their corresponding descriptions in both manuscript and supplementary material.

Impacts: The changes slightly affect the quantitative results but do not alter the conclusions that the refined aerodynamic roughness length improves WRF performance over high-roughness regions. A comparison and description of the main figures and tables before and after the correction are provided in the document below.

We sincerely apologize for the oversight and appreciate your understanding.

Kun Yang, on behalf of all co-authors

**Figure and Table Comparisons**

- 1. Before the correction,  $z_0$  values were estimated for 1,805 stations, which increased to 1,837 stations after correction. Consequently, the number of stations shown in Figs. 1 and 2 has increased. Some of the stations show slight numerical variations in  $z_0$  values. The urban-rural classification types remain unchanged, but the counts for some categories have slightly varied.
- 2. Because the number of  $z_0$  estimates increased, we retrained the random forest model, resulting in updates to Fig. 3.
- 3. The annual mean  $z_0$  in Fig. 4 differs slightly between the two versions, with absolute differences in most regions being less than 0.2 in  $\ln(z_0)$  (Fig. R1), which is considered acceptable.

**Figure R1.** Difference in the annual mean  $\ln(z_0)$  before and after the correction, i.e., the corrected  $\ln(z_{0\_RFR})$  (Fig. 4a) minus the previous one.

4. Figures 5-7 present the evaluation of simulated wind speeds, and the numerical differences are minor. We mainly list Fig. 6 below, which compare model performance using 10 m wind speeds from CMA stations. After the time alignment correction of station observations, the temporal correlation coefficients of wind speed simulations have improved noticeably (Fig. 6d).

Figure 6 (previous). (a) Comparisons of mean 10-m wind speed in April between the simulations using  $z_{0\_Default}$ ,  $z_{0\_Peng}$ , and  $z_{0\_RFR}$  versus observations from CMA stations. All points (grey circles and purple crosses) represent the 753 CMA stations within the d02 domain available for comparison, while the purple crosses represent the 155 stations utilized for independent validation, which were not used in training the  $z_{0\_RFR}$  model. The corresponding wind speed means, R, and RMSE of all stations are also indicated. (b) Distribution of the 155 independent CMA stations (black stars). Colored shaded areas represent TSD. (c) Comparison of daily mean 10-m wind speed between simulations and observations from 753 CMA stations. (d) Statistical metrics comparing simulated and observed 10-m wind speeds, including temporal and spatial R, MAB, and RMSE.

**Figure 6 (correction).** (a) Comparisons of mean 10-m wind speed in April between the simulations using  $z_{0\_Default}$ ,  $z_{0\_Peng}$ , and  $z_{0\_RFR}$  versus observations from CMA stations. All points (grey circles and purple crosses) represent the 753 CMA stations within the d02 domain available for comparison, while the purple crosses represent the 148 stations utilized for independent validation, which were not used in training the  $z_{0\_RFR}$  model. The corresponding wind speed means, R, and RMSE of all stations are also indicated. (b) Distribution of the 148 independent CMA stations (black stars). Colored shaded areas represent TSD. (c) Comparison of daily mean 10-m wind speed between simulations and observations from 753 CMA stations. (d) Statistical metrics comparing simulated and observed 10-m wind speeds, including temporal and spatial R, MAB, and RMSE.

5. In Table 1, the mean wind speeds before and after correction differ only slightly. However, since the percentage reduction in wind speed error is sensitive to small changes, the corresponding values show a noticeable decrease. Nevertheless, the improvement in wind speed due to the updated  $z_0$  remains highly significant."

**Table 1 (previous).** Mean 10-m wind speed at 753 CMA stations and mean 100-m wind speed at 50 anemometer towers from simulations and observations. Simulations were performed using  $z_{0\_Default}$ ,  $z_{0\_Peng}$ , and  $z_{0\_RFR}$ . Also shown is the percentage reduction in wind speed error (*PRE*) achieved by  $z_{0\_RFR}$  relative toz0\\_Default and  $z_{0\_Peng}$ .

|                            | $Z_{0\_Default}$ | $Z_{0\_Peng}$ | $Z_{0\_RFR}$ | Observations |
|----------------------------|------------------|---------------|--------------|--------------|
| Mean 10-m wind speed (m/s) | 2.97             | 2.89          | 2.17         | 2.08         |

| PRE in 10-m wind speed (%)  | 89.9% | 88.9% | -    | -    |
|-----------------------------|-------|-------|------|------|
| Mean 100-m wind speed (m/s) | 7.10  | 7.27  | 6.38 | 6.26 |
| PRE in 100-m wind speed (%) | 85.7% | 88.1% | -    | -    |

**Table 1 (correction).** Mean 10-m wind speed at 753 CMA stations and mean 100-m wind speed at 50 anemometer towers from simulations and observations. Simulations were performed using  $z_{0\_Default}$ ,  $z_{0\_Peng}$ , and  $z_{0\_RFR}$ . Also shown is the percentage reduction in wind speed error (*PRE*) achieved by  $z_{0\_RFR}$  relative to  $z_{0\_Default}$  and  $z_{0\_Peng}$ .

|                             | $Z_{0\_Default}$ | $z_{0\_Peng}$ | $z_{0\_RFR}$ | Observations |
|-----------------------------|------------------|---------------|--------------|--------------|
| Mean 10-m wind speed (m/s)  | 2.97             | 2.90          | 2.26         | 2.08         |
| PRE in 10-m wind speed (%)  | 79.8%            | 78.0%         | -            | -            |
| Mean 100-m wind speed (m/s) | 7.09             | 7.29          | 6.50         | 6.26         |
| PRE in 100-m wind speed (%) | 71.1%            | 76.7%         | -            | -            |

---

## Referee Report (RR1)

**Summary**

This manuscript provides a new dataset of roughness z0 values focused on "built up" areas in China, which covers a large percentage of the country. Their method for generating the $z_0$ values is by first using meteorological stations and ERA5 to derive site specific z0 values, then using random forests to calculate a gridded z0 dataset based on six different inputs (slop variation, terrain standard deviation, percent tree cover, leaf area index, normalized difference vegetation index, and urban-rural classification). They then test their z0 dataset using a nested WRF setup and find improved agreement with observations compared to the default z0 values and also a recent z0 dataset from another study. Overall, the paper is well written, and the method is very clear. I think the method is a nice approach that others could use for their own datasets; however, I do feel that a few minor revisions need to be made before the paper should be published.

**General Comments**

1. The manuscript's main goal is improving z0 values in "built up" areas. I'm assuming this is anywhere there is infrastructure, which could be urban, suburban, residential, etc. However, the random forest regression analysis in Fig. 3 demonstrates that that the urban rural classification is actually only the 4th most important input parameter in determining z0. The two most important inputs are actually just the terrain itself. This, to me, indicates that the most likely reason models are underestimating wind speed closer to the surface is just because the terrain is under resolved with coarser grid spacing. Perhaps there are other reasons, regardless, it is not clear in the manuscript that "land use changes such as urbanization", as stated in the abstract, are the reason z0 data is not accurate. I believe that the feature importance figure requires additional discussion, and these findings are worthy enough of being restated in the conclusions and the abstract.

   The feature importance figure is probably the most important finding that I took away after reading the manuscript. The authors certainly constructed a better z0 database, but the reason other z0 databases are wrong seems to not necessarily be because of urbanization.

   I think this comment is inline with the "(3) Lack of Resolution-Dependent $z_0$ Consideration" comment from another reviewer. The authors stated in their response "the horizontal resolution of ERA5 does not affect the estimated $z_0$ values at individual stations." This is true, but then the gridded z0 dataset is resolution dependent because the high resolution (100 m?) SRTM data is used as an input to the

RFR but then the simulations are done at a much coarser resolution. Whether the other reviewer is satisfied with the authors response is obviously up to the other reviewer but, in my opinion, the most straightforward way to look at the resolution-dependence would be to add an additional domain at finer resolution and see if there is improvement even with the default z0 values. Or, considering that the authors already ran a multiscale setup, they could compare results between d01 and d02.

2. Some figures use ln(z0) as the parameter that's being shown: Fig. 4, Fig. 8, Fig 1b and 1c. This is not very intuitive for the reader since z0 itself has units of meters and a physical meaning. These figures would be much clearer if the z0 value was shown and then the axes or colorbar were logged.

3. Along those same lines, any time lnz0 is used it should be ln(z0), this would improve the clarity of the manuscript significantly.

4. I had a similar comment as one of the other reviewers regarding the circular logic in using ERA5 data to derive z0. I believe other readers would question this, as well. The findings in the supplementary confirm that there is improved agreement with the NCEP data. I think those findings should be included more as an appendix in the manuscript rather than supplementary material. I believe the additional discussion section that was added could probably move to the appendix along with the relevant supplementary material, but that is ultimately up to the authors to decide.

5. Lastly, I think the error metrics should all be defined with their equations in Section 2.5. For example, when the authors restate MBP in the captions, the equations being inline make them difficult to read.

**Specific comments**

- Line 46: remove "of"
- Line 210: change "confirms the reasonableness of the z0_optimal" to "confirms that z0_optimal is reasonable".
- Fig 3a does not have a fully white or transparent background. Additionally, Fig. 3d doesn't need a grid
- Line 358: I'd suggest rewriting to avoid using representativeness

---

## Referee Report (RR2)

**Review of "Improvement of near-surface wind speed modeling through refined aerodynamic roughness length in built-up regions: implementation and validation in the Weather Research and Forecasting (WRF) model version 4.0"**

This manuscript addresses an important challenge in meteorology and wind energy applications. The authors propose a cost-effective method to estimate aerodynamic roughness length at CMA weather stations by reconciling discrepancies between ERA5 reanalysis and surface observations, then extend these estimates to a gridded dataset using Random Forest Regression. The refined dataset is implemented in WRF and compared with default and alternative z0 datasets. Results show improvements in 10-m and 100-m wind simulations. The study is timely, practical, and demonstrates methodological robustness across reanalyses and meteorological conditions.

I appreciate the novelty and practicality of the proposed approach. The validation experiments convincingly show that the new dataset reduces biases in WRF simulations. To further strengthen the paper, I encourage the authors to enhance the methods section and provide additional clarifications.

1. If I understand correctly, the study reconstructs ERA5 100 m wind speeds from the log-law (Eq. 2–3) using 10 m winds and $z_{0,\mathrm{ERA5}}$. However, ERA5 also provides 100 m wind speed as a native output. It would be useful to explicitly state this, clarify why the reconstructed values were preferred. It would also be helpful to evaluate the potential differences between the log-law–derived 100 m winds and native ERA5 100 m winds, and discuss any implications for the derived $z_0$.

2. The method assumes that stability impacts are identical in ERA5 and CMA observations and thus neglects stability corrections. While this simplification is reasonable for efficiency and consistency, ERA5 simulated stability itself may be biased, potentially introducing additional uncertainty. Previous studies evaluating vertical interpolation methods (e.g., the NREL report by Duplyakin et al., 2021, https://www.nrel.gov/docs/fy21osti/78412.pdf) found that simple neutral log-law interpolation often performs best in U.S. wind resource assessments. Referencing this evidence would strengthen the justification for the assumption.

3. ERA5 ingests a wide range of observations, including surface and upper-air data. While 10 m or 100 m winds are not necessarily assimilated directly, the assimilation of pressure, temperature, and upper-level winds improves boundary-layer structure and indirectly benefits surface-layer winds. A short discussion of this point would emphasize the credibility of ERA5 data as the reference in the proposed method.

4. The Introduction focuses strongly on the importance of $z_0$ in dense urban areas. However, Fig. 2d shows that the dataset also covers lower-density built-up regions

and natural surfaces such as residential areas and woodlands. Extend the introduction to cover natural vegetation would strengthen the scope.

5. Since natural regions are included, vegetation phenology (e.g., foliage status) could influence $z_0$. The October case therefore provides a meaningful seasonal contrast to April and would be better discussed in the main text rather than only in the Supplementary Material.

6. The feature importance results (Fig. 3e) are interesting, especially the dominance of topographic predictors relative to vegetation metrics. It is somewhat surprising that LAI appears less important, given that leaf phenology can strongly influence roughness. This may result from collinearity with NDVI, which can bias feature importance rankings, or from averaging across all regions, thereby masking deciduous-seasonal effects. An extended discussion of this result would be helpful. Alternatively, if the authors feel the analysis adds little value, it could be streamlined.

---

## Author Response (AR2)

**Responses to the Reviewers**

We would like to express our sincere gratitude to all reviewers for their insightful and constructive comments. Their valuable suggestions have greatly contributed to improving the clarity and overall quality of our manuscript. Based on their feedback, we have carefully revised and supplemented the manuscript accordingly.

During revision, we realized that several concerns from all three reviewers stemmed from our use of the term "built-up areas" to describe the study area, which may have been misleading. In the original manuscript, this term referred to the following categories from the urban-rural classification dataset of Li et al. (2023): Urban center, Urban landscape, Densely clustered towns, Sparsely clustered towns, Dense villages, Sparse villages, Isolated villages, Residential croplands, Populated croplands, Residential rangelands, and Residential woodlands. However, we realized that "built-up areas" is commonly understood as regions with intensive infrastructure, such as urban, suburban, or residential zones, which does not fully reflect our definition, since some "residential" classes include croplands and woodlands. To more accurately capture the physical characteristics of these surfaces, we have replaced "built-up areas" with "**high-roughness surface areas**" throughout the manuscript, except where the term specifically denotes built-up areas. This revised term better represents regions characterized by large surface roughness, including both urban areas and landscapes with tall vegetation.

The detailed responses to each comment raised by the reviewers are presented in the following sections. The responses are highlighted in blue, and the corresponding revisions in the manuscript are marked in red. We sincerely hope that these revisions address all concerns and meet the reviewers' expectations.

**Reviewer #1:**

**General Comments:**

This manuscript provides a new dataset of roughness $z_0$ values focused on "built up" areas in China, which covers a large percentage of the country. Their method for generating the $z_0$ values is by first using meteorological stations and ERA5 to derive site specific $z_0$ values, then using random forests to calculate a gridded $z_0$ dataset based on six different inputs (slop variation, terrain standard deviation, percent tree cover, leaf area index, normalized difference vegetation index, and urban-rural classification). They then test their $z_0$ dataset using a nested WRF setup and find improved agreement with observations compared to the default $z_0$ values and also a recent $z_0$ dataset from another study. Overall, the paper is well written, and the method is very clear. I think the method is a nice approach that others could use for their own datasets; however, I do feel that a few minor revisions need to be made before the paper should be published.

**Response:** We sincerely appreciate your positive feedback and constructive comments. Your suggestions have been carefully considered and have significantly contributed to the improvement of our manuscript. Our detailed point-by-point responses are provided below.

**Specific comments:**

1. The manuscript's main goal is improving $z_0$ values in "built up" areas. I'm assuming this is anywhere there is infrastructure, which could be urban, suburban, residential, etc. However, the random forest regression analysis in Fig. 3 demonstrates that that the urban rural classification is actually only the 4th most important input parameter in determining $z_0$. The two most important inputs are actually just the terrain itself. This, to me, indicates that the most likely reason models are underestimating wind speed closer to the surface is just because the terrain is under resolved with coarser grid spacing. Perhaps there are other reasons, regardless, it is not clear in the manuscript that "land use changes such as urbanization", as stated in the abstract, are the reason $z_0$

data is not accurate. I believe that the feature importance figure requires additional discussion, and these findings are worthy enough of being restated in the conclusions and the abstract.

The feature importance figure is probably the most important finding that I took away after reading the manuscript. The authors certainly constructed a better $z_0$ database, but the reason other $z_0$ databases are wrong seems to not necessarily be because of urbanization.

I think this comment is in line with the "(3) Lack of Resolution-Dependent $z_0$ Consideration" comment from another reviewer. The authors stated in their response "the horizontal resolution of ERA5 does not affect the estimated $z_0$ values at individual stations." This is true, but then the gridded $z_0$ dataset is resolution dependent because the high resolution (100 m?) SRTM data is used as an input to the RFR but then the simulations are done at a much coarser resolution. Whether the other reviewer is satisfied with the authors response is obviously up to the other reviewer but, in my opinion, the most straightforward way to look at the resolution dependence would be to add an additional domain at finer resolution and see if there is improvement even with the default z0 values. Or, considering that the authors already ran a multiscale setup, they could compare results between d01 and d02.

**Response:** Thank you for your valuable comment. We would first like to clarify a terminology revision. Based on the issues raised by you and other reviewers, we realized that our previous use of the term *"built-up areas"* was not sufficiently accurate. In the original manuscript, this term referred to 11 categories from the urban-rural classification (*URC*) dataset of Li et al. (2023): Urban center, Urban landscape, Densely clustered towns, Sparsely clustered towns, Dense villages, Sparse villages, Isolated villages, Residential croplands, Populated croplands, Residential rangelands, and Residential woodlands. However, we acknowledge that *"built-up areas"* is generally understood as regions characterized by intensive infrastructure (e.g., urban, suburban, or residential zones), which does not fully align with our definition, particularly because some "residential" categories include croplands and woodlands. To more accurately

describe the physical characteristics of the underlying surfaces considered in this study, we have replaced "built-up areas" with "high-roughness surface areas" throughout the manuscript. In the following, we will respond in detail to your comments concerning (1) the feature importance, and (2) the resolution dependence.

(1) Feature importance

We appreciate your insightful comment regarding the relatively low importance of the *URC* variable in determining $z_0$. In our study, the *URC* variable primarily served to define and constrain the study areas rather than act as a dominant predictor of $z_0$. In other words, the inclusion of *URC* was intended to ensure that the random forest model was trained only over high-roughness surface areas. It should be noted that aerodynamic roughness characteristics related to surface morphology, such as vegetation density and infrastructure distribution, are already inherently captured in the CMA station wind observations. In other words, $z_0$ derived from observed wind speeds already reflects the influence of high-roughness surfaces, while *URC* serves mainly to delineate the spatial coverage of $z_0$ across different *URC* categories.

The relatively low importance of the *URC* variable is due to the relatively small differences in $z_0$ among different *URC* classes. $z_0$ is governed primarily by the morphological height and density of surface roughness elements, whereas the *URC* is not defined by these attributes; instead, it integrates global land-cover and population data (Li et al., 2023), making it weakly sensitive to $z_0$. Moreover, even in categories of Urban center and Urban landscape, there remains non-negligible tree cover, mean tree fractions of approximately 10% and 11%, respectively (Fig. R1). This lowers *URC*'s ranking in feature-importance analyses. To better capture the influence of roughness elements, more detailed surface parameters, such as building height and building density, would be helpful. Once such data are widely accessible, they should be incorporated to further improve the accuracy of $z_0$ estimates.

[Figure]

**Figure R1.** Mean percent tree cover (*PTC*) for each *URC*. The numerical labels on the x-axis represent the *URC* codes, with the specific *URC* types annotated on the bars.

(2) Resolution dependence

The above explanation regarding feature importance also indicates that, even though terrain features rank among the top two important features in the random forest model, it cannot be concluded that the correction of wind speed overestimation in the WRF is due to finer terrain resolution. In fact, the fundamental reason is that the $z_0$ derived from CMA observations over high-roughness surface can better reflect the influence of these areas. The gridded $z_0$ data we produced can effectively improve wind speed simulations because the training truth values are more accurate and can better represent the actual surface roughness. This is the central point of our study: we propose a low-cost approach to substantially supplement $z_0$ truth values. The purpose of $z_0$ data production and wind simulation validation is primarily to demonstrate the feasibility and effectiveness of this $z_0$ estimation method. Therefore, the model's overestimation of wind speed still stems from insufficient consideration of the influence of high-roughness surfaces.

We also thank you for the suggestion to demonstrate the resolution dependence by comparing simulations at different resolutions. Following this idea, we compared the simulation results between the d01 and d02 domains. Figure R2 shows the simulated

wind speeds using different $z_0$ datasets with the coarse-resolution (0.09° for d01) and fine-resolution (0.03° for d02) simulations. Overall, the mean wind speeds at both 10 m and 100 m heights are close between the two domains across all $z_0$ cases. Similar to Figs. 6a/6d and 7a/7d in the manuscript, we further analyzed the performance of the coarse-resolution simulations.

As shown in Figs. R3-R4, $z_{0\_RFR}$ still leads to a significant improvement in the near-surface wind speed simulations, even at coarse resolution. However, the improvement magnitude does not show a consistent dependence on model resolution. For example, when using the $z_{0\_Default}$ or $z_{0\_RFR}$, the finer-resolution simulations generally yield lower *RMSE* values of 10 m wind speed compared with the coarser-resolution simulations. In contrast, when using the $z_{0\_Peng}$, the d02 simulation produces a higher *RMSE* than d01 (Fig. 6a and Fig. R3a). Overall, these results suggest that although the $z_0$ dataset was developed at a 0.01° resolution, it is well suited for mesoscale simulations at kilometer-level resolutions.

[Figure]

**Figure R2.** Comparison of mean 10-m (a) and 100-m (b) wind speeds in April between the coarse-resolution (0.09°; d01) and fine-resolution (0.03°; d02) simulations using $z_{0\_Default}$, $z_{0\_Peng}$, and $z_{0\_RFR}$. Each point corresponds to (a) a CMA station or (b) an

anemometer tower located within the d02 domain. The overall mean wind speed across all observation sites is also shown.

[Figure]

**Figure R3.** (a) Comparison of mean 10-m wind speeds in April between the coarse-resolution (0.09°; d01) simulations using $z_{0\_Default}$, $z_{0\_Peng}$, and $z_{0\_RFR}$ and observations from CMA stations. All points (grey circles and purple crosses) represent the 753 CMA stations within the d02 domain available for comparison, while the purple crosses represent the 155 stations utilized for independent validation, which were not used in training the $z_{0\_RFR}$ model. (b) Comparison of mean 100-m wind speeds in April between the coarse-resolution (0.09°; d01) simulations using $z_{0\_Default}$, $z_{0\_Peng}$, and $z_{0\_RFR}$ and observations from anemometer towers. The corresponding wind speed means, *R*, and *RMSE* of all stations are also indicated.

[Figure]

**Figure R4.** Statistical comparison of the coarse-resolution (0.09°; d01) simulations and observations within the d02 domain. (a) 10-m wind speeds from 753 CMA stations, and (b) 100-m wind speeds from 50 anemometer towers. Temporal and spatial *R*, *MAB*, and *RMSE* are included.

Based on the responses above, and to provide greater clarity for the readers, we have made the following additions to the manuscript:

(1) In the revised manuscript, we have added an explanation of *URC* importance in lines 277-288: "The *URC* ranks only fourth in feature importance. This ranking should not be interpreted as implying that land use or urbanization is insignificant. Rather, in our framework, *URC* is used mainly to delineate the study domain and to ensure that the RFR algorithm is applied only to high-roughness surface areas. The aerodynamic effects of high-roughness elements, such as tall vegetation, buildings, and other infrastructure, are already embedded in the wind observations from CMA stations. As a result, the influence of these roughness elements is directly reflected in the $z_0$ values themselves, rather than being captured by the *URC*. Essentially, *URC* is not defined in terms of the morphological height and density of roughness elements; instead, it is derived from global land-cover and population data (Li, X. et al., 2023), and is therefore weakly sensitive to $z_0$. For example, in categories of Urban center and Urban landscape,

there remains non-negligible tree cover, mean tree fractions of approximately 10% and 11%, respectively (Fig. S1b). This lowers *URC*'s ranking in feature-importance analyses. To better capture the influence of roughness elements, more detailed surface parameters, such as building height and building density, would be helpful. Once such data are widely accessible, they should be incorporated to further improve the accuracy of $z_0$ estimates." Figure R1 has been added to the Supplementary Materials as Fig. S1b.

(2) We have added a discussion on the resolution dependence that "Therefore, $z_0$ values estimated from 10-m wind observations are reasonably representative at ~100 m-1 km scales, making the generation of 0.01° gridded $z_0$ datasets for use in mesoscale simulations both appropriate and justified, with no evident resolution dependence observed. We compared simulation results at different resolutions. Leveraging the nested modeling setup used in this study, the d01 domain with a 0.09° resolution was treated as the coarse-resolution simulation, while d02 at 0.03° served as the fine-resolution simulation. The results show that, even at the coarser resolution, our gridded $z_0$ dataset provides a clear advantage and substantially improves near-surface wind speed simulations (Fig. S8 and S9)." in lines 417-422 of the revised manuscript, and added Figs. R2-R3 to the supplementary material as Figs. S8-S9.

(3) In Abstract, raw expression that "In built-up areas, surface roughness has been substantially altered by land use changes such as urbanization. However, many numerical models assign $z_0$ values based on vegetation cover types neglecting urban effects. This has resulted in a lack of reliable $z_0$ data in built-up regions." has been replaced by "In high-roughness surface areas, surface roughness has been substantially altered by land use changes such as urbanization. However, many numerical models still assign long-standing and fixed $z_0$ based on traditional land cover types, neither accounting for shifts in land cover nor updating class-specific $z_0$, leaving $z_0$ values in high-roughness surface regions outdated and unreliable."

2. Some figures use $\ln(z_0)$ as the parameter that's being shown: Fig. 4, Fig. 8, Fig 1b and 3c. This is not very intuitive for the reader since $z_0$ itself has units of meters and

a physical meaning. These figures would be much clearer if the $z_0$ value was shown and then the axes or colorbar were logged.

**Response:** Thank you for your valuable suggestion. In the revised manuscript, we have made the following changes accordingly. The axes in Figs. 3c and 8a-b have been changed to logarithmic scales, and Fig. 4 now presents the distribution of $z_0$ instead of $\ln(z_0)$, with an adjusted colorbar to better highlight the differences among datasets. However, we have retained $\ln(z_0)$ in Figs. 1b and 8c, because $z_0$ affects the wind profile in a logarithmic manner based on the Monin-Obukhov similarity theory. Therefore, comparing $\ln(z_0)$ values provides a more direct and physically meaningful interpretation of relative differences. For example, $z_0$ values of 0.01 m and 0.02 m differ by 0.01, while 0.001 m and 0.002 m differ by 0.001, but both pairs have the same $\ln(z_0)$ difference ($-0.69$), indicating comparable aerodynamic impacts.

3. Along those same lines, any time $\ln z_0$ is used it should be $\ln(z_0)$, this would improve the clarity of the manuscript significantly.

**Response:** Thank you for your suggestion. We have carefully checked the entire manuscript and revised all occurrences of "$\ln z_0$" to the correct format "$\ln(z_0)$" to improve clarity and consistency.

4. I had a similar comment as one of the other reviewers regarding the circular logic in using ERA5 data to derive $z_0$. I believe other readers would question this, as well. The findings in the supplementary confirm that there is improved agreement with the NCEP data. I think those findings should be included more as an appendix in the manuscript rather than supplementary material. I believe the additional discussion section that was added could probably move to the appendix along with the relevant supplementary material, but that is ultimately up to the authors to decide.

**Response:** We appreciate your comment. We fully agree that the cross-validation using NCEP data is important to demonstrate the robustness of the refined $z_0$ dataset. However, since these results serve as supporting evidence rather than the main focus of this study, we consider it is more appropriate to keep them in the supplementary

material. We have ensured that the main text clearly refers to these supplementary results for readers who wish to examine the validation details.

5. Lastly, I think the error metrics should all be defined with their equations in Section 2.5. For example, when the authors restate MBP in the captions, the equations being inline make them difficult to read.

**Response:** Thank you for your valuable suggestion. In the revised manuscript, the definitions of all error metrics have been moved to Section 2.5. The definitions that were previously included in figure and table captions have been removed to improve readability and clarity.

6. Line 46: remove "of"

**Response:** Thank you for pointing this out. The word "of" has been removed as suggested in line 48 of the revised manuscript.

7. Line 210: change "confirms the reasonableness of the $z_{0\_optimal}$" to "confirms that $z_{0\_optimal}$ is reasonable".

**Response:** Thank you for the suggestion. We have revised the text accordingly to "supports that $z_{0\_optimal}$ is reasonable" in lines 246-247 of the revised manuscript.

8. Fig 3a does not have a fully white or transparent background. Additionally, Fig. 3d doesn't need a grid.

**Response:** Thank you for the comment. In the revised manuscript, we have updated Fig. 3a to have a fully white background, and the grid in Fig. 3d has been removed as suggested.

9. Line 358: I'd suggest rewriting to avoid using representativeness.

**Response:** Thank you for the suggestion. We have revised the sentence to avoid using "representativeness." The sentence in lines 413-415 of the revised manuscript now reads: "Moreover, the agreement between ERA5- and WRF-derived $z_0$ values suggests that the spatial extent represented by the estimated site-level $z_0$ values is not

determined by the resolution of the reanalysis or simulation dataset used, but rather by the measurement height of wind observations at the stations."

**Reference**

Li, X., Yu, L. and Chen, X.: New insights into urbanization based on global mapping and analysis of human settlements in the rural-urban continuum, Land, 12, 1607, doi:10.3390/land12081607, 2023.

**Reviewer #2:**

**General Comments:**

This manuscript introduced an approach to estimate roughness length at CMA weather stations ($z_{0\_CMA}$) that minimizes differences in 100-m wind speed ($u_{100}$) between ERA5 ($u_{100\_ERA5}$) and CMA ($u_{100\_CMA}$) stations (Equations 2 and 3), using the wind profile described by Monin-Obukhov similarity theory flux-profile relationship in neutral conditions (Equation 1). They assumed differences in near-surface wind speed between ERA5 and CMA stations are mainly due to $z_0$ and the influence of $z_0$ diminishes with height (i.e., $u_{100\_ERA5} \sim u_{100\_CMA}$) (assumption 1), and the impact of atmospheric stability on wind speed is identical between ERA5 and CMA (assumption 2) (Lines 122-125). The estimated station-wise $z_0$ ($z_{0\_CMA}$) was then used to derive a gridded $z_0$ dataset using a random forest regression algorithm, i.e., $z_{0\_RFR}$, which improved near-surface wind simulations in the WRF model compared to those simulations with other static $z_0$ values.

This manuscript is well written and organized. This manuscript not only provides a method to estimate roughness length at measurement stations but also suggests a potential way to provide a gridded dataset that can be applied to numerical simulations, which well fits the scope of this journal. I do not 100% agree with the authors on the two assumptions that they used to derive $z_{0\_CMA}$ (assumptions 1 and 2 above). I think impacts of the validity of these two assumptions are topics to be discussed and further studied (it would be good if this issue is briefly discussed in the manuscript), but they don't need to be addressed in the current manuscript. Below I have several minor comments and suggestions.

**Response:** We sincerely appreciate your positive comments regarding the manuscript's clarity, organization, and the proposed methodology. We also thank you for highlighting the two assumptions used to derive $z_{0\_CMA}$. We acknowledge that the two assumptions cannot be fully verified with the available data. However, adopting the $z_{0\_CMA}$ values derived under them markedly improves WRF-simulated wind speeds

compared with those using default $z_0$ values. Therefore, from a practical modeling perspective, these assumptions represent reasonable and effective approximation. We also provide further explanation and discussion of their rationality from the following aspects.

For Assumption 1, the original intent is: "(1) the near-surface wind speed difference between ERA5 and CMA is primarily attributed to $z_0$, and the influence of $z_0$ diminishes with height. Consequently, at higher levels within the near-surface layer, the wind speed from ERA5 reanalysis is considered comparable to that from observations;" And we selected 100 m as the analysis height for two reasons: (1) ERA5 provides near-surface wind speed only at 100 m, and (2) several anemometer towers we collected also include wind speed observations at 100 m, which can indirectly support the validity of the assumption. Specifically, the estimated $z_{0\_CMA}$ values under this assumption are mainly concentrated in eastern China, while those at most western stations are difficult to estimate accurately. This pattern is generally consistent with the observation that the bias between ERA5 and tower-measured 100-m wind speeds is much smaller in eastern China than in the west (Figs. 1c and 2a in the manuscript).

Additionally, we conducted a sensitivity experiment on the choice of reference height, re-estimating $z_{0\_CMA}$ using 150 m and 200 m (Fig. R1). As shown in Fig. R1 a and c, the annual-mean $z_{0\_CMA}$ derived from 150 m or 200 m is broadly consistent with that obtained at 100 m: for 88.6% (150 m) and 87.3% (200 m) of stations, the absolute difference from the 100-m-based estimate is less than 0.5. Stations that are more sensitive to the reference height may be influenced by local terrain complexity, as suggested in Fig. R1 b and d.

In short, both the model-improvement outcomes and the sensitivity-test results indicate that this assumption is reasonable and practically useful. Nevertheless, some deviations in the simulated wind speeds still exist. If more multi-height observational data become available in the future, it would allow testing alternative reference height assumptions and obtaining more precise $z_0$ estimates.

[Figure]

**Figure R1.** Sensitivity of the reference-height choice in Assumption 1 for $z_{0\_CMA}$ estimation. (a, c) Distributions of station counts by the absolute difference in annual-mean $z_{0\_CMA}$ values between estimates using 150 m (a) or 200 m (c) and those using 100 m. (b, d) Boxplots of *TSD* across bins of the absolute difference in annual-mean derived $ln(z_0)$. (b, d) Boxplots of *TSD* versus binned absolute differences in annual-mean $z_{0\_CMA}$, computed relative to the 100-m–based estimate for 150 m (b) and 200 m (d).

For Assumption 2, we assume that the impact of atmospheric stability on wind speed is identical for both ERA5 and CMA stations, allowing us to neglect explicit stability corrections when estimating $z_{0\_CMA}$. In the Monin-Obukhov similarity framework, the stability correction term is generally smaller in magnitude than the logarithmic term. Moreover, when estimating $z_{0\_CMA}$, the stability correction term appears in both the numerator and the denominator of the governing equation. Therefore, their effects can be considered to approximately cancel each other out. This simplification is reasonable from both an efficiency and consistency standpoint, as also supported by the validation of simulated wind speeds. Additionally, Duplyakin et al. (2021) have shown that incorporating stability corrections into vertical interpolation of wind profile does not

necessarily improve accuracy. They compared multiple interpolation schemes in U.S. wind resource assessments and found that neutral log-law method performed comparably to stability-corrected version. This also supports that such an approximate treatment seems feasible and a widely adopted simplification.

Based on the above, we have added a brief discussion about the two assumptions in Section 4. Discussion of the revised manuscript (lines 448-465): "The two assumptions used in the $z_0$ estimation are also discussed. Although these assumptions cannot be fully verified with the available data, they are pragmatically motivated and indirectly supported by the improved performance of wind-speed simulations using the resulting $z_0$ estimates. Assumption 1 posits that the near-surface wind-speed discrepancy between ERA5 reanalysis and CMA observations is dominated by $z_0$ and that the influence of $z_0$ weakens with height, making ERA5 winds at higher levels within the surface layer comparable to observations. This is partly supported by the spatial pattern of estimated $z_0$ (denser over eastern China, where 100-m wind-speed biases between ERA5 reanalysis and anemometer tower observations are smaller (Figs. 1c and 2a)) and by a sensitivity test on the reference height (Figs. S11a and S11c). When re-estimating annual-mean $z_{0\_CMA}$ at 150 m and 200 m, 88.6% and 87.3% of stations, respectively, show an absolute difference from the 100-m–based estimate below 0.5, indicating broad consistency across heights. A minority of stations exhibit larger deviation, which may be influenced by local terrain complexity (Figs. S11b and S11d). Assumption 2 treats the effects of atmospheric stability on wind speed as effectively similar in ERA5 and at CMA sites, allowing us to omit explicit stability corrections in estimating $z_{0\_CMA}$. This simplification enhances methodological consistency and computational efficiency, and it is indirectly supported by the validation of simulated winds. Moreover, prior work has shown that neutral log-law method can perform comparably to stability-corrected scheme for vertical interpolation in U.S. wind-resource assessments (Duplyakin et al., 2021), suggesting that such an approximate treatment seems feasible and a widely adopted simplification. Overall, although neither assumption can be fully verified with the presently available data, their practical applicability is evidenced by improved WRF

wind-speed simulations. Future work, ideally leveraging multi-height wind profile observations and coincident stability metrics could further test these assumptions, yield more precise $z_0$ estimates." Figure. R1 has been added to the supplementary material as Fig. S11.

We have also carefully considered your following suggestions, which have substantially improved the clarity and rigor of the manuscript. Detailed, point-by-point responses to each comment are provided below.

**Specific comments:**

1. Lines 92-93, "variance of the slope": Could you clarify what data you used to derive this variable with its spatial resolution (e.g., 3 arcsec SRTM)?

**Response:** Thank you for the question. The variance of the slope ($\overline{\theta^2}$) used in our study was sourced from the dataset accompanying the turbulent orographic form drag scheme in WRF (Zhou et al., 2018). This dataset was processed from the global 30″ GMTED2010 digital elevation model (Danielson & Gesch, 2011; ~1 km nominal spacing at the equator). To clarify this in the manuscript, we added the following sentence that "We obtained $\overline{\theta^2}$ from the dataset accompanying the turbulent orographic form drag scheme in WRF (Zhou et al., 2018), which was processed from the global 30″ GMTED2010 digital elevation model (Danielson & Gesch, 2011)." in lines 102-104 of the revised manuscript.

2. Line 100, "$z_0$ dataset at a spatial resolution of 0.01°×0.01°": Why did you select this spatial resolution?

**Response:** Thank you for raising this point. We chose a spatial resolution of 0.01°×0.01° for two reasons: (1) Our objective is to develop a $z_0$ dataset suitable for mesoscale modeling with kilometer-level resolutions; (2) The "rule of thumb" for near-surface wind observations is that the horizontal representativeness is roughly 10-100 times the measurement height. Since our $z_0$ truth values are inferred from 10-m CMA winds, the representative footprint is on the order of ~100m-1 km, which is well matched by a

0.01° grid. Thus, constructing a 0.01° gridded $z_0$ dataset is both appropriate and justified, which was explained in lines 89-91 and 417-418.

However, this does not imply that the wind-speed simulations necessarily exhibit any resolution dependence. To address this (also raised by Reviewer 1), we tested the performance at different model resolutions using our nested setup: the outer domain (d01, 0.09°) as a coarse-resolution case and the inner domain (d02, 0.03°) as a fine-resolution case. Across both nests, the $z_0$ dataset delivers clear and consistent improvements in near-surface wind simulations (Figs. R2-R3), indicating no evident resolution dependence within the mesoscale range considered. These results support that a 0.01° dataset is suitable and effective for kilometer-scale WRF applications. We have added the supporting multi-resolution comparison to the revised manuscript in lines 417-422: "Therefore, $z_0$ values estimated from 10-m wind observations are reasonably representative at ~100 m-1 km scales, making the generation of 0.01° gridded $z_0$ datasets for use in mesoscale simulations both appropriate and justified, with no evident resolution dependence observed. We compared simulation results at different resolutions. Leveraging the nested modeling setup used in this study, the d01 domain with a 0.09° resolution was treated as the coarse-resolution simulation, while d02 at 0.03° served as the fine-resolution simulation. The results show that, even at the coarser resolution, our gridded $z_0$ dataset provides a clear advantage and substantially improves near-surface wind speed simulations (Fig. S8 and S9)."

[Figure]

**Figure R2.** (a) Comparison of mean 10-m wind speeds in April between the coarse-resolution (0.09°; d01) simulations using $z_{0\_Default}$, $z_{0\_Peng}$, and $z_{0\_RFR}$ and observations from CMA stations. All points (grey circles and purple crosses) represent the 753 CMA stations within the d02 domain available for comparison, while the purple crosses represent the 155 stations utilized for independent validation, which were not used in training the $z_{0\_RFR}$ model. (b) Comparison of mean 100-m wind speeds in April between the coarse-resolution (0.09°; d01) simulations using $z_{0\_Default}$, $z_{0\_Peng}$, and $z_{0\_RFR}$ and observations from anemometer towers. The corresponding wind speed means, *R*, and *RMSE* of all stations are also indicated.

[Figure]

**Figure R3.** Statistical comparison of the coarse-resolution (0.09°; d01) simulations and observations within the d02 domain. (a) 10-m wind speeds from 753 CMA stations, and (b) 100-m wind speeds from 50 anemometer towers. Temporal and spatial *R*, *MAB*, and *RMSE* are included.

3. Line 206, "all stations are situated in build-up areas": Not "all" stations seem to be situated in build-up areas. Figure 2c shows there are some stations at croplands categories. Am I missing something?

**Response:** Thank you for pointing this out. You are correct that not all stations lie in what is commonly called "built-up areas." In our original manuscript, we used "built-up areas" as shorthand for the urban-rural classifications of Li et al. (2023) that define our study area, including Urban center, Urban landscape, Densely clustered towns, Sparsely clustered towns, Dense villages, Sparse villages, Isolated villages, Residential croplands, Populated croplands, Residential rangelands, and Residential woodlands. As you pointed, several of these classes encompass croplands and woodlands, even those labeled "Populated" or "Residential", so "built-up" is not an accurate descriptor. To more accurately reflect the surface characteristics relevant to our study, we have replaced "built-up areas" with "high-roughness surface areas" throughout the manuscript. This term is intended to encompass both urban/town environments and

landscapes with tall vegetation (e.g., residential croplands and woodlands) that imply relatively large $z_0$. These wording changes are terminological clarifications only and do not affect our analysis or conclusions.

4. Lines 210-211, "the robust consistency in the relationship between $z_0$ and wind speed confirmed the reasonableness of $z_{0\_optimal}$": I think the robust consistency in the relationship between $z_{0\_optimal}$ and wind speed is an expected outcome because you used equation 1, which relates $z_0$ and wind speed, to derive $z_{0\_CMA}$, which was in turn used to derive $z_{0\_optimal}$. In that regard, I'm not sure how the robustness relationship between $z_0$ and wind speed can be related to the reasonableness of $z_{0\_optimal}$.

**Response:** Thank you for your careful reading and for raising this point about Fig. 1d. We agree that, if one looks only at Eq. (1) (in the manuscript) at a single time step, a strong $z_0$-wind-speed relationship is expected and should not be taken as proof of the validity of $z_{0\_optimal}$. Our intention with Fig. 1d was not to use that relationship as an independent validation, but rather as a consistency check: to verify that, under our statistical aggregation procedure, the derived $z_{0\_optimal}$ varies with wind-speed bias in the expected manner.

Specifically, for each station we estimated instantaneous $\ln(z_0)$ at each hour using Eqs. (2)-(3) (in the manuscript), and then defined the station's monthly $\ln(z_{0\_optimal})$ as the median of all hourly values within that month. Because the $\ln(z_{0\_optimal})$ is a cross-time statistic (median) rather than a direct substitution at any single time, the instantaneous implication from Eqs. (2)-(3), namely, "if $u_{10\_ERA5} > u_{10\_CMA}$ then $z_{0\_ERA5} < z_{0\_CMA}$", does not necessarily persist after aggregation. We therefore relate binned difference of $\ln(z_0)$ to the mean bias percentage of 10-m wind speed in Fig. 1d to check whether the monotonic, theory-consistent pattern remains after aggregation, rather than to "prove" $z_{0\_optimal}$. The reasonableness of $z_{0\_optimal}$ is assessed through independent validation by comparing 10- and 100-m wind speeds simulated based on $z_{0\_optimal}$ against observations (Figs. 6-7 and Table. 1 in the manuscript).

To avoid any impression of circular validation, we have revised the relevant passages in the manuscript accordingly to: "Additionally, as a consistency check, we examined how the difference in $\ln(z_0)$ covaries with the 10-m wind-speed bias between ERA5 reanalysis and station observations." in lines 231-232 and "Because $\ln(z_{0\_optimal})$ is defined as a monthly median of hourly $\ln(z_0)$, this cross-time statistic does not trivially inherit the instantaneous relationship implied by Equations (1)-(3). The monotonic, theory-consistent pattern observed in the binned $\ln(z_0)$ difference versus wind-speed *MBP* therefore serves as a post-aggregation consistency check, rather than as proof. Accordingly, the robust consistency in the relationship between $z_0$ and wind speed preliminarily supports that $z_{0\_optimal}$ is reasonable, and suggests that improving $z_0$ values over high-roughness surface areas in numerical models could significantly enhance wind speed simulation accuracy. The validity of $z_{0\_optimal}$ will be assessed via independent validation by comparing simulated wind speeds with observations (Section 3.3)." In Lines 243-249.

5. Line 215, Figure 1: It is hard to read Figures 1a-1c. I think using color scales that are more consistent with numerical values (e.g., bluish/reddish colors for negative/positive values) would help.

**Response:** Thank you for the helpful suggestion. We have updated Figs. 1a-1c in the revised manuscript.

6. Line 230, "categores": "categories".

**Response:** Thank you for pointing this out. The typo has been corrected from "categores" to "categories".

7. Lines 333-334, "the resulting gridded $z_0$ dataset significantly reduces uncertainties ~ particularly over relatively flat built-up areas": The WRF model has various urban canopy model (UCM) options, which parameterize effects of urban topography by updating surface drag etc. This includes for example, a single-layer UCM (WRF UCM option 1), a building effect parameterization (BEP; WRF UCM option 2), etc. Considering the impact of the updated roughness length dataset is mainly over urban

areas, could you explain the advantage of using the updated dataset instead of using an UCM? Also, could you compare the impact of $z_{0\_optimal}$ with the impact of using UCMs?

**Response:** Thank you for the thoughtful question. It likely stems from our potentially misleading use of the term "built-up". We have revised the terminology, replacing "built-up areas" with "high-roughness surface areas". Accordingly, our updated roughness-length dataset applies not only to urban areas. Additionally, our study targets mesoscale wind simulations at kilometer-scale grid spacing (~1-10 km). We are less familiar with UCMs, but our review confirms that UCMs and our $z_0$ dataset are designed for different purposes and are suited for different simulation scales. UCMs were conceived to operate at ~0.5-1 km grid spacing to bridge mesoscale forecasting (~105 m) with microscale transport/dispersion (~100 m) models (Tewari et al., 2006; Chen et al., 2010), and they are most commonly applied at ~1 km resolution (Lian et al., 2018; Salamanca et al., 2018; Wang et al., 2021). At our target kilometer-scale resolutions, our $z_0$ dataset offers a more practical and effective path for improving near-surface wind simulations for the following three reasons. First, regarding parameter dependence, our approach bypasses the need for land-use classification, sources of error that directly impact UCMs performance. Second, in terms of computational cost, our method is highly efficient, avoiding the expensive prognostic calculations required by UCMs such as BEP. Third, for model configuration, our solution offers greater flexibility and portability, as it does not impose the specific PBL and LSM combinations that some UCMs demand.

Therefore, a direct comparison of wind speed simulations using our $z_0$ dataset and UCMs within the scope of this work is not undertaken for two primary reasons. First, as our expertise and research focus lie in mesoscale simulation, we are less familiar with the specific application of UCMs. More importantly, these approaches are designed for fundamentally different purposes. Our $z_0$ dataset is tailored for efficient mesoscale wind simulation at kilometer-scale resolutions, whereas UCMs are the preferred and more specialized tool for investigating detailed urban canopy effects, such as building-induced drag and urban thermodynamics.

Based on the above, we have added a brief discussion of the applicable spatial resolution of our $z_0$ dataset in lines 423-429 of the revised manuscript: "However, for simulations at ~1 km resolution and finer, such as urban-scale wind modelling, our $z_0$ dataset cannot fully capture urban heterogeneity, because it did not incorporate key morphological parameters (e.g., building height and density) to distinguish between different urban forms. Therefore, an urban canopy model (UCM) would be a more appropriate choice. UCMs were conceived to operate at ~0.5-1 km grid spacing to bridge mesoscale forecasting (~$10^5$ m) with microscale transport/dispersion (~$10^0$ m) models (Tewari et al., 2006; Chen et al., 2010), and they have been widely applied and validated in subsequent urban studies (Lian et al., 2018; Salamanca et al., 2018; Wang et al., 2021). Therefore, our $z_0$ data are suitable and effective for mesoscale simulations at kilometer-level resolutions."

**References**

Chen, F., Kusaka, H., Bornstein, R., Ching, J., Grimmond, C. S. B., Grossman-Clarke, S., Loridan, T., Manning, K. W., Martilli, A., Miao, S., Sailor, D., Salamanca, F. P., Taha, H., Tewari, M., Wang, X., Wyszogrodzki, A. A., and Zhang, C.: The integrated WRF/urban modelling system: development, evaluation, and applications to urban environmental problems, Int. J. Climatol., 31, 273–288, doi:10.1002/joc.2158, 2011.

Danielson, J. J., and Gesch, D. B.: Global multi-resolution terrain elevation data 2010 (GMTED2010), US Geological Survey, No. 2011-1073, doi:10.3133/ofr20111073, 2011.

Duplyakin, D., Zisman, S., Phillips, C., and Tinnesand, H.: Bias characterization, vertical interpolation, and horizontal interpolation for distributed wind siting using mesoscale wind resource estimates, National Renewable Energy Laboratory (NREL), Golden, CO, USA, NREL/TP-2C00-78412, doi:10.2172/1760659, 2021.

Li, X., Yu, L. and Chen, X.: New insights into urbanization based on global mapping and analysis of human settlements in the rural-urban continuum, Land, 12, 1607, doi:10.3390/land12081607, 2023.

Lian, J., Wu, L., Bréon, F.-M., Broquet, G., Vautard, R., Zaccheo, T. S., Dobler, J., and Ciais, P.: Evaluation of the WRF-UCM mesoscale model and ECMWF global operational forecasts over the Paris region in the prospect of tracer atmospheric transport modeling, Elem. Sci. Anth., 6, 64, doi:10.1525/elementa.319, 2018.

Salamanca, F., Zhang, Y., Barlage, M., Chen, F., Mahalov, A., and Miao, S.: Evaluation of the WRF-urban modeling system coupled to Noah and Noah-MP land surface models over a semiarid urban environment, J. Geophys. Res.-Atmos., 123, 2387–2408, doi:10.1002/2018JD028377, 2018.

Tewari, M., Chen, F., and Kusaka, H.: Implementation and evaluation of a single-layer urban canopy model in WRF/Noah, In: Proceedings of the WRF Users' Workshop, NCAR, Boulder, CO, USA, 2006.

Wang, J. and Hu, X.-M.: Evaluating the performance of WRF urban schemes and PBL schemes over Dallas–Fort Worth during a dry summer and a wet summer, J. Appl. Meteorol. Climatol., 60, 779–798, doi:10.1175/JAMC-D-19-0195.1, 2021.

Zhou, X., Yang, K. and Wang, Y.: Implementation of a turbulent orographic form drag scheme in WRF and its application to the Tibetan Plateau, Clim. Dynam., 50, 2443-2455, doi:10.1007/s00382-017-3677-y, 2018.

**Reviewer #3:**

**General Comments:**

This manuscript addresses an important challenge in meteorology and wind energy applications. The authors propose a cost-effective method to estimate aerodynamic roughness length at CMA weather stations by reconciling discrepancies between ERA5 reanalysis and surface observations, then extend these estimates to a gridded dataset using Random Forest Regression. The refined dataset is implemented in WRF and compared with default and alternative z0 datasets. Results show improvements in 10-m and 100-m wind simulations. The study is timely, practical, and demonstrates methodological robustness across reanalysis and meteorological conditions.

I appreciate the novelty and practicality of the proposed approach. The validation experiments convincingly show that the new dataset reduces biases in WRF simulations. To further strengthen the paper, I encourage the authors to enhance the methods section and provide additional clarifications.

**Response:** We would like to express our sincere gratitude for your positive feedback and insightful comments and suggestions. These have significantly enhanced the quality of our manuscript. We have carefully considered all your points. In the following sections, we provide a detailed response to each of your comments.

**Specific comments:**

1. If I understand correctly, the study reconstructs ERA5 100 m wind speeds from the log-law (Eq. 2-3) using 10 m winds and $z_{0\_ERA5}$. However, ERA5 also provides 100 m wind speed as a native output. It would be useful to explicitly state this, clarify why the reconstructed values were preferred. It would also be helpful to evaluate the potential differences between the log-law–derived 100 m winds and native ERA5 100 m winds, and discuss any implications for the derived $z_0$.

**Response:** We thank you for raising this important point. Indeed, ERA5 provides native 100-m winds. Our choice to use the log-law-reconstructed 100-m winds from $u_{10\_ERA5}$

and $z_{0\_ERA5}$ rather than the native ERA5 100-m winds can be justified from two following perspectives.

First, our $\ln(z_0)$ estimation relies on two assumptions that warrant equating Eqs. (2) and (3) for the optimization. Assumption 2 permits neglecting stability terms solely when Eqs. (2) and (3) are set equal, meaning ERA5 and CMA have identical stability and the effect cancels to some extent. This does not justify we can ignore the stability for wind speed calculations. However, the native ERA5 100-m wind inherently includes stability effects. Therefore, directly pairing native ERA5 100-m winds with our CMA log-law construction would amplify the error in the derived $\ln(z_0)$.

Second, our $z_0$ estimation method requires only 10-m wind speeds and $z_0$ from reanalysis, together with 10-m wind speeds from observations. It does not rely on reanalysis winds at other heights, which makes the approach low-cost and easily extensible. For Assumption 1, the original intent is: "(1) the near-surface wind speed difference between ERA5 and CMA is primarily attributed to $z_0$, and the influence of $z_0$ diminishes with height. Consequently, at higher levels within the near-surface layer, the wind speed from ERA5 reanalysis is considered comparable to that from observations;" And we selected 100 m as the analysis height for two reasons: (1) ERA5 provides near-surface wind speed only at 100 m, and (2) several anemometer towers we collected also include wind speed observations at 100 m, which can indirectly support the validity of the assumption. Specifically, the estimated $z_{0\_CMA}$ values under this assumption are mainly concentrated in eastern China, while those at most western stations are difficult to estimate accurately. This pattern is generally consistent with the observation that the bias between ERA5 and tower-measured 100-m wind speeds is much smaller in eastern China than in the west (Figs. 1c and 2a in the manuscript). Additionally, to illustrate the dependence on height selection, we conducted a sensitivity experiment, re-estimating $z_{0\_CMA}$ using 150 m and 200 m (Fig. R1). As shown in Fig. R1 a and c, the annual-mean $z_{0\_CMA}$ derived from 150 m or 200 m is broadly consistent with that obtained at 100 m: for 88.6% (150 m) and 87.3% (200 m) of stations, the absolute difference from the 100-m-based estimate is less than 0.5.

Stations that are more sensitive to the reference height may be influenced by local terrain complexity, as suggested in Fig. R1b and d.

[Figure]

**Figure R1.** Sensitivity of the reference-height choice in Assumption 1 for $z_{0\_CMA}$ estimation. (a, c) Distributions of station counts by the absolute difference in annual-mean derived $z_{0\_CMA}$ between estimates using 150 m (a) or 200 m (c) and those using 100 m. (b, d) Boxplots of *TSD* across bins of the absolute difference in annual-mean derived $ln(z_0)$. (b, d) Boxplots of *TSD* versus binned absolute differences in annual-mean $z_{0\_CMA}$, computed relative to the 100-m–based estimate for 150 m (b) and 200 m (d).

In summary, lines 142-150 of the revised manuscript now clearly explain why we prefer the reconstructed values and the advantages of this approach: "Actually, ERA5 provides native 100-m winds, but here we use log-law–reconstructed 100-m winds from $u_{10\_ERA5}$ and $z_{0\_ERA5}$ instead. The reason is that the $z_{0\_CMA}$ is derived under the assumption that stability-correction term is neglected. This means that the 100-m wind speeds in Equations (2) and (3) are both calculated without considering stability effects. However, the native ERA5 100-m wind field inherently embeds model-diagnosed stability influences. Therefore, directly pairing native ERA5 100-m winds with our

CMA log-law construction would amplify the error in the derived $\ln(z_0)$. In addition, the reconstruction offers two practical advantages. First, it requires fewer variables and a more transparent linkage, relying only on 10-m wind speeds and $z_0$ from reanalysis, together with 10-m wind speeds from observations; Second, our results indicate that the $z_0$ estimates are not particularly sensitive to the choice of reference height (see Section 4. Discussion), so there is no need to use native reanalysis winds at heights other than 10 m."

Details of the sensitivity to the choice of reference height are reported in lines 448-457: "The two assumptions used in the $z_0$ estimation are also discussed. Although these assumptions cannot be fully verified with the available data, they are pragmatically motivated and indirectly supported by the improved performance of wind-speed simulations using the resulting $z_0$ estimates. Assumption 1 posits that the near-surface wind-speed discrepancy between ERA5 reanalysis and CMA observations is dominated by $z_0$ and that the influence of $z_0$ weakens with height, making ERA5 winds at higher levels within the surface layer comparable to observations. This is partly supported by the spatial pattern of estimated $z_0$ (denser over eastern China, where 100-m wind-speed biases between ERA5 reanalysis and anemometer tower observations are smaller (Figs. 1c and 2a)) and by a sensitivity test on the reference height (Figs. S11a and S11c). When re-estimating annual-mean $z_{0\_CMA}$ at 150 m and 200 m, 88.6% and 87.3% of stations, respectively, show an absolute difference from the 100-m–based estimate below 0.5, indicating broad consistency across heights. A minority of stations exhibit larger deviation, which may be influenced by local terrain complexity (Figs. S11b and S11d)."

2. The method assumes that stability impacts are identical in ERA5 and CMA observations and thus neglects stability corrections. While this simplification is reasonable for efficiency and consistency, ERA5 simulated stability itself may be biased, potentially introducing additional uncertainty. Previous studies evaluating vertical interpolation methods (e.g., the NREL report by Duplyakin et al., 2021, https://www.nrel.gov/docs/fy21osti/78412.pdf) found that simple neutral log-law

interpolation often performs best in U.S. wind resource assessments. Referencing this evidence would strengthen the justification for the assumption.

**Response:** We appreciate your helpful suggestion. To strengthen the justification for Assumption 2, we have added a citation to Duplyakin et al. (2021) in lines 457-462: "Assumption 2 treats the effects of atmospheric stability on wind speed as effectively similar in ERA5 and at CMA sites, allowing us to omit explicit stability corrections in estimating $z_{0\_CMA}$. This simplification enhances methodological consistency and computational efficiency, and it is indirectly supported by the validation of simulated winds. Moreover, prior work has shown that neutral log-law method can perform comparably to stability-corrected scheme for vertical interpolation in U.S. wind-resource assessments (Duplyakin et al., 2021), suggesting that such an approximate treatment seems feasible and a widely adopted simplification."

3. ERA5 ingests a wide range of observations, including surface and upper-air data. While 10 m or 100 m winds are not necessarily assimilated directly, the assimilation of pressure, temperature, and upper-level winds improves boundary-layer structure and indirectly benefits surface-layer winds. A short discussion of this point would emphasize the credibility of ERA5 data as the reference in the proposed method.

**Response:** Thank you for the suggestion. We have added a brief discussion to emphasize the credibility of ERA5 in lines 409-412 of the revised manuscript: "Meanwhile, although 10-m and 100-m winds over lands are not assimilated directly in ERA5, its 4D-Var system ingests a wide range of surface and upper-air observations that constrain boundary-layer structure and indirectly improve near-surface winds; this strengthens the credibility of using ERA5 as the reference field (Hersbach et al., 2020)."

4. The Introduction focuses strongly on the importance of $z_0$. in dense urban areas. However, Fig. 2d shows that the dataset also covers lower-density built-up regions and natural surfaces such as residential areas and woodlands. Extend the introduction to cover natural vegetation would strengthen the scope.

**Response:** Thank you for the helpful suggestion. First, we have replaced the term "built-up areas" with "high-roughness surface areas" throughout to reflect that our study spans both built-up regions and vegetated landscapes. Second, we refined the study-area expression and incorporated a concise description of natural-vegetation roughness effects. The specific changes are as follows:

Lines 31-33: "With the rapid advancement of urbanization and industrialization, human activities and energy use are increasingly concentrated along the settlement-landscape continuum (Liu et al., 2014), particularly in high-roughness areas such as built-up zones and inhabited vegetated landscapes."

Lines 48-52: "Specifically, most models, such as the widely used ECMWF Reanalysis v5 (ERA5), determine $z_0$ with long-standing and fixed values based on traditional land cover types. Such treatment fails to reflect the impact of transitions between surface types and changes in roughness elements within the same type, particularly the complexity of urban structures, thereby posing significant challenges for accurate wind speed simulation and prediction over high-roughness surface areas (Wang et al., 2024)."

Lines 52-54: "Numerous studies have demonstrated that the changes of $z_0$, caused by land use changes, particularly urbanization and industrialization, as well as deforestation and afforestation, significantly impacted wind speed."

Lines 59-62: "A similar mechanism operated in Canada. At Sudbury Airport (Ontario), 10-m wind speeds declined by ~34% during 1975-1995 mainly due to reforestation-induced increases in surface roughness (Tanentzap et al., 2007). These findings highlight the need to refine $z_0$ in models by incorporating the effects of high-roughness surface areas across urban-town settings and tall-vegetation landscapes."

Lines 64-65: "Winckler et al. (2019) showed that roughness changes are a primary control on deforestation's biogeophysical effects, notably surface temperature responses."

5. Since natural regions are included, vegetation phenology (e.g., foliage status) could influence $z_0$. The October case therefore provides a meaningful seasonal contrast to

April and would be better discussed in the main text rather than only in the Supplementary Material.

**Response:** We appreciate your point that vegetation phenology can modulate $z_0$ and that October offers a meaningful seasonal contrast to April. Because the main text is already dense and our primary contribution is the method development and wind improvement, we keep the figures about October in the Supplementary Material. However, we have added a comparison of performance across April and October in lines 386-390 of the revised manuscript, as follows: "Station-wise correlations increase and errors decrease to a similar extent in both months, and the daily time series likewise show closer tracking of peaks and lulls. Taken together, these results further reinforce the reliability and applicability of the proposed $z_0$ estimation under varying meteorological conditions. They also indicate that although phenology-driven changes in canopy structure and seasonal circulation modulate wind speeds, the performance advantage of the proposed $z_0$ is not diminished."

6. The feature importance results (Fig. 3e) are interesting, especially the dominance of topographic predictors relative to vegetation metrics. It is somewhat surprising that *LAI* appears less important, given that leaf phenology can strongly influence roughness. This may result from collinearity with *NDVI*, which can bias feature importance rankings, or from averaging across all regions, thereby masking deciduous-seasonal effects. An extended discussion of this result would be helpful. Alternatively, if the authors feel the analysis adds little value, it could be streamlined.

**Response:** Thank you for your insightful question. $z_0$ is primarily determined by the characteristic height of surface roughness elements, particularly their relief. As a result, topographic features rank among the most important factors. For vegetation characteristics, PTC not only captures the horizontal distribution of vegetation density but also serves as a proxy for the presence of tall roughness elements. However, LAI mainly reflects vegetation density, making it relatively less critical.

To assess the role of collinearity, we first note that *LAI* and *NDVI* are moderately correlated across the 1,805 stations ($R = 0.72$). And We conducted a post-hoc

permutation test on the test subset by randomly shuffling *NDVI*, which has reduced $R^2$ by 3.2% and increased *RMSE* by 6.2%. Taking the RFR model used in this study as the baseline, we conducted an ablation study by retraining two control models with the same data splits and hyperparameters, removing *LAI* in one and *NDVI* in the other. In both cases, test performance was broadly similar to the baseline (Table R1). When *NDVI* was removed, the impurity importance of *LAI* still ranked below topographic predictors and *PTC* (Fig. R2). Therefore, these results indicate that the relatively low impurity-based importance of *LAI* is not primarily driven by collinearity of *LAI* and *NDVI*.

We have added a discussion of *LAI* importance in lines 273-277 of the revised manuscript: "$z_0$ is primarily controlled by the characteristic height of surface roughness elements, particularly their relief. Consequently, topographic features rank among the most influential factors. For vegetation-related features, *PTC* not only reflects the horizontal distribution of vegetation density but also serves as a proxy for the presence of tall roughness elements. By contrast, *LAI* mainly represents vegetation density, making it relatively less critical. Although *LAI* is strongly correlated with *NDVI* ($R = 0.72$), its low importance is not driven by this collinearity."

**Table R1.** Test-subset performance of the Random Forest Regression (RFR) under baseline and ablation settings. The baseline refers to the raw RFR model used in our study trained with all features. Two ablation cases are retrained with identical splits and hyperparameters, excluding *LAI* in one case and *NDVI* in the other.

|  | Baseline | *NDVI*-excluded | *LAI*-excluded |
|---|---|---|---|
| *R* | 0.90 | 0.90 | 0.90 |
| *RMSE* | 0.11 | 0.11 | 0.12 |

[Figure]

[Figure]

**Figure R2.** Importance scores of feature variables under *NDVI*-excluded case and *LAI*-excluded case.

**References**

Duplyakin, D., Zisman, S., Phillips, C., and Tinnesand, H.: Bias characterization, vertical interpolation, and horizontal interpolation for distributed wind siting using mesoscale wind resource estimates, National Renewable Energy Laboratory (NREL), Golden, CO, USA, NREL/TP-2C00-78412, doi:10.2172/1760659, 2021.

Hersbach, H., Bell, B., Berrisford, P., Hirahara, S., Horányi, A., Muñoz-Sabater, J., Nicolas, J., Peubey, C., Radu, R., Schepers, D., Simmons, A., Soci, C., Abdalla, S., Abellan, X., Balsamo, G., Bechtold, P., Biavati, G., Bidlot, J., Bonavita, M., De Chiara, G., Dahlgren, P., Dee, D., Diamantakis, M., Dragani, R., Flemming, J., Forbes, R., Fuentes, M., Geer, A., Haimberger, L., Healy, S., Hogan, R. J., Hólm, E., Janisková, M., Keeley, S., Laloyaux, P., Lopez, P., Lupu, C., Radnoti, G., de Rosnay, P., Rozum, I., Vamborg, F., Villaume, S., and Thépaut, J.-N.: The ERA5 global reanalysis, Q. J. R. Meteorol. Soc., 146, 1999-2049, doi:10.1002/qj.3803, 2020.

Li, X., Yu, L. and Chen, X.: New insights into urbanization based on global mapping and analysis of human settlements in the rural-urban continuum, Land, 12, 1607, doi:10.3390/land12081607, 2023.

Liu, Z., He, C., Zhou, Y. and Wu, J.: How much of the world's land has been urbanized, really? A hierarchical framework for avoiding confusion, Landsc. Ecol., 29, 763-771, doi:10.1007/s10980-014-0034-y, 2014.

Tanentzap, A. J., Taylor, P. A., Yan, N. D., and Salmon, J. R.: On Sudbury-area wind speeds—a tale of forest regeneration. Journal of applied meteorology and climatology, 46(10), 1645-1654, doi:10.1175/JAM2552.1, 2007.

Wang, J., Yang, K., Yuan, L., Liu, J., Peng, Z., Ren, Z. and Zhou, X.: Deducing aerodynamic roughness length from abundant anemometer tower data to inform wind resource modeling, Geophys. Res. Lett., 51, e2024GL111056, doi:10.1029/2024GL111056, 2024.

Winckler, J., Reick, C. H., Bright, R. M. and Pongratz, J.: Importance of surface roughness for the local biogeophysical effects of deforestation, J. Geophys. Res.-Atmos., 124, 8605-8618, doi:10.1029/2018JD030127, 2019.

---

## Author Response (AR3)

**Responses to the Reviewers**

We would like to express our sincere gratitude to the editor and all reviewers for their constructive and insightful comments. Their valuable suggestions have been extremely helpful in improving both the clarity and scientific quality of our manuscript. In this revision, we have carefully addressed every comment raised and provided detailed, point-by-point responses in the following sections.

In addition, during a workflow check, we identified a time misalignment in the processing of meteorological station observations used for aerodynamic roughness length estimation and model evaluation. This issue has now been corrected. After reprocessing the data, we found that the correction caused minor numerical differences in several evaluation metrics and figures, but the conclusions remain unchanged.

Our replies to reviewers' comments are highlighted in blue, and the corresponding revisions in the manuscript are marked in red. We hope that these revisions satisfactorily address all concerns and meet the expectations of the editor and reviewers.

**Editor:**

Line 26: "a latest gridded $z_0$ dataset": change to "a latest gridded $z_0$ dataset from Peng et al. (2022)"

Line 196: "The cumulus parameterization scheme was exclusively activated in the d02 domain": the d02 resolution is about 3km. Do you mean no CU schemes for D02? Or CU only applies to d01?

**Response:** Thank you for your suggestions. We have revised both parts accordingly. Regarding the clarification on the cumulus parameterization scheme, we apologize for the typos in the original manuscript. The scheme was activated only in the d01 domain and switched off in the d02 domain. The text has been revised accordingly to: "The cumulus parameterization scheme was activated only in the d01 domain and switched off in the d02 domain."

**Reviewer #1:**

Thank you to the authors for responding to all of my comments. I appreciate all of the revisions they made and believe this manuscript to be publishable.

I noticed a couple of small typos that I assume will eventually be handled but figured to mention them here:

- line 49: spell out ECMWF.

- line 142: remove "actually".

- Section 2.5: the overbar for some variables includes the subscript. I believe the correct notation is for the overbar to be only over the variable itself and not the subscript.

**Response:** We thank you for your positive feedback and for pointing out these typos. We have implemented all the suggested corrections:

On line 49, "ECMWF" has been spelled out.

On line 142, the word "actually" has been removed.

In Section 2.5, the overbar notation has been corrected to appear only over the variable itself, not the subscript.